# Surface remodeling and inversion of cell-matrix interactions underlie community recognition and dispersal in *Vibrio cholerae* biofilms

Alexis Moreau[1], Danh T. Nguyen[2], Alexander J. Hinbest[3], Anthony Zamora[4], Ranjuna Weerasekera [3], Katherine Matej [1], Xuening Zhou [5], Sandra Sanchez[6], Ignacio Rodriguez Brenes[4], Jung-Shen Benny Tai[1], Carey D. Nadell [7,8], Wai-Leung Ng[6], Vernita Gordon [5,9], Natalia L. Komarova[10], Rich Olson [3], Ying Li [2] & Jing Yan [1,11] ✉

Biofilms are ubiquitous surface-associated bacterial communities embedded in an extracellular matrix. It is commonly assumed that biofilm cells are glued together by the matrix; however, how the specific biochemistry of matrix components affects the cell-matrix interactions and how these interactions vary during biofilm growth remain unclear. Here, we investigate cell-matrix interactions in *Vibrio cholerae*, the causative agent of cholera. We combine genetics, microscopy, simulations, and biochemical analyses to show that *V. cholerae* cells are not attracted to the main matrix component (*Vibrio* polysaccharide, VPS), but can be attached to each other and to the VPS network through surface-associated VPS and crosslinks formed by the protein Bap1. Downregulation of VPS production and surface trimming by the polysaccharide lyase RbmB cause surface remodeling as biofilms age, shifting the nature of cell-matrix interactions from attractive to repulsive and facilitating cell dispersal as aggregated groups. Our results shed light on the dynamics of diverse cell-matrix interactions as drivers of biofilm development.

Biofilms are surface-associated bacterial communities embedded in an extracellular matrix, which varies between species but is generally a mixture of exopolysaccharide (EPS), accessory proteins, and other components[1,2]. Biofilm growth is a widespread element of microbial life in nature, but they can also obstruct municipal systems and cause fouling of industrial surfaces[3]. Biofilm formation increases antibiotic tolerance and creates high densities of cells at sites of infection, which can lead to worse outcomes for biofilm-related disease[4]. Despite extensive research on the genetic and regulatory networks governing biofilm formation, the detailed biophysical mechanism underlying the

[1]Department of Molecular, Cellular and Developmental Biology, Yale University, New Haven, CT, USA. [2]Department of Mechanical Engineering, University of Wisconsin-Madison, Madison, WI, USA. [3]Department of Molecular Biology and Biochemistry, Molecular Biophysics Program, Wesleyan University, Middletown, CT, USA. [4]Department of Mathematics, University of California Irvine, Irvine, CA, USA. [5]Interdisciplinary Life Sciences Graduate Program, Center for Nonlinear Dynamics, The University of Texas at Austin, Austin, TX, USA. [6]Department of Molecular Biology and Microbiology, Tufts University School of Medicine, Boston, MA, USA. [7]Department of Biological Sciences, Dartmouth Colleague, Hanover, NH, USA. [8]Department of Microbiology and Immunology, Geisel school of Medicine at Dartmouth, Lebanon, NH, USA. [9]Department of Physics, LaMontagne Center for Infectious Disease, The University of Texas at Austin, Austin, TX, USA. [10]Department of Mathematics, University of California San Diego, La Jolla, CA, USA. [11]Quantitative Biology Institute, Yale University, New Haven, CT, USA. ✉e-mail: jing.yan@yale.edu

cell-matrix interaction remains poorly understood. Conventional wisdom holds that this matrix glues the biofilm-dwelling cells together into cell aggregates[5-7]; however, this picture has yet to keep up with our increasing understanding of the biochemistry of the complex extracellular matrix. In certain species such as *Staphylococcus epidermidis*, the matrix polymers are positively charged and therefore intrinsically attractive to the negatively-charged bacterial cell surface[8]. However, in the more general case where the exopolysaccharides are neutral or negatively charged, electrostatics alone cannot be the sole explanation for the biofilm architecture.

The intuitive picture of attractive cell-matrix interaction is also not always easy to reconcile with a number of observations from the biofilm research community. It is reported that some biofilms can exclude other species and even non-matrix-producing mutants of the same clonal strain background[9-11]: if the existing matrix is inherently attractive to the bacterial surface, how exactly can we explain these patterns of cell exclusion from existing biofilm structures? Another challenge faced by biofilm-forming cells is how to efficiently escape the matrix during dispersal upon nutrient limitation[12]. While essential for biofilm structural integrity, EPS creates a physical barrier for dispersing cells. Specific enzymes have been found to digest the matrix; however, increasing evidence suggests that dispersal can take place on short time scales (< 30 min) with incomplete or no degradation of the matrix[13-15], suggesting other unknown biophysical factors facilitating dispersal.

Here, we addressed these questions using the model biofilm-former *Vibrio cholerae* (*Vc*), the etiological agent of the diarrheal disease cholera[16,17]. The biofilm matrix of *Vc* is primarily composed of Vibrio polysaccharide (VPS), a secreted polysaccharide consisting of a unique negatively charged tetrasaccharide repeating unit[18,19]. VPS plays the major role in maintaining the structural integrity of *Vc* biofilms[20] and all accessory proteins rely on VPS to form a functional matrix[16,21]. Among the accessory proteins, RbmA has been shown to adhere biofilm-residing cells to each other[22-24], whereas Bap1 and RbmC contribute to cell-to-surface adhesion as well as to the formation of envelope structures surrounding *Vc* biofilm clusters[21,25,26]. The crystal structures of these proteins have been solved recently[22,23,27-30], although the biochemical details of how they interact with VPS are still unclear. In addition, a putative polysaccharide lyase RbmB has been shown to be required for proper biofilm dispersal[14,25,31], a process in which biofilm-dwelling cells transition back to a planktonic state and leave an existing biofilm, usually triggered by nutrient limitation[12]. RbmB has been shown to degrade VPS in vitro[32], but exactly how it promotes biofilm dispersal is unclear due to a limited biochemical and structural understanding.

## Results

### VPS is not attractive to *Vibrio cholerae* cells

Our first goal was to identify the nature of the interaction between VPS and *Vc* cells. Extracellular polymers can facilitate cell aggregation through two main mechanisms: bridging or depletion[33]. Bridging occurs when the polymers have an attractive interaction to the cell surface, allowing them to simultaneously adsorb onto multiple cells while bringing these cells into proximity. Depletion, on the other hand, is driven by entropy and takes place when non-attractive or repulsive polymers and cells coexist[34-36]. In this case, cells aggregate to allow the polymer molecules to explore more space, hence maximizing the configurational entropy and minimizing the free energy of the entire system (Fig. S1). The morphology of aggregates produced by bridging is qualitatively different from the morphology of aggregates produced by depletion[33]: Bridging typically results in disordered and loose aggregates; in contrast, depletion compacts rod-shaped cells into a parallel, space-filling arrangement. Moreover, the scaling of the concentration of polymer required for aggregation with the number density of cells has opposite signs for the two mechanisms – positive with bridging, and negative with depletion. For the bridging

mechanism, more polymers are needed to cause aggregation as cell concentration increases; consequently, the boundary separating the aggregated and non-aggregated phases on a two-dimensional (2D) phase diagram of polymer concentration versus cell density exhibits a positive slope. In contrast, depletion attraction is promoted by the crowding of polymers and cells; therefore, fewer polymers are needed to aggregate higher concentrations of cells, generating a phase boundary with a negative slope.

To observe the aggregate morphology and map the 2D phase diagram of polymer concentration versus cell concentration for *Vc* and its matrix, we purified VPS and visualized the polymer-induced aggregation of a Δ*rbmA*Δ*bap1*Δ*rbmC*Δ*vpsL*Δ*pomA* (hitherto abbreviated as 5Δ) strain. This is a nonmotile mutant (due to the deletion of flagellar motor protein PomA[37]) unable to produce VPS or any major matrix proteins. This choice minimizes confounding factors due to bacterial motility and the production of matrix during the aggregation assay. We used several non-native, well-characterized polymers with different charges as controls: positively-charged polymers including chitosan and poly-L-lysine (PLL), negatively-charged polymers including polystyrene sulfonate (PSS), and neutral polymers including dextrans of different molecular weights. The aggregation patterns differed significantly between the polymers: chitosan and PLL induced disordered aggregates and phase boundaries with a positive slope (Fig. 1a and Fig. S2). This pattern indicates aggregation by bridging of negatively-charged cell surfaces by the positively-charged polymers[38]. On the other hand, when mixed with PSS or dextran, *Vc* cells form aggregates of parallelly aligned cells with phase boundaries with a negative slope (Figs. S1 and S2), characteristic of depletion-attraction. Establishing these controls enabled us to unequivocally test the interaction between purified VPS (pVPS) and 5Δ cells.

We find that pVPS induced cell aggregation in a manner similar to the repulsive polymers (Fig. 1a), evidenced by a negative slope in the phase diagram and the parallel alignment of cells within aggregates; these are characteristics of depletion-attraction.

### Depletion-aggregation happens spontaneously in *Vc* culture

To determine whether depletion attraction can drive spontaneous aggregation during cell culture, we first used a strain capable of producing VPS but not the major matrix proteins (Δ*rbmA*Δ*bap1*Δ*rbmC*, abbreviated as Δ*ABC*). The absence of all major matrix proteins leads to a loose, expanded network of VPS[39], although the physiochemical properties of the network and the interaction between VPS molecules (entanglement, self-association, etc.) remain uncharacterized. This strain also carries a point mutation in *vpvC*, a diguanylate cyclase that synthesizes cyclic diguanylate (c-di-GMP)[40], the major second messenger controlling the transition from the planktonic to biofilm state[41]. In *Vc*, it is well established that c-di-GMP upregulates the production of VPS and matrix proteins, through binding and activating transcriptional regulators including VpsT and VpsR[16,17]. The *vpvC*[W240R] mutation was introduced to increase the intracellular level of c-di-GMP and the baseline biofilm matrix production rate[40]; all biofilm assays below use this genetic background.

Depletion-aggregation indeed occurs spontaneously after ~10 hours when Δ*ABC* cells were allowed to grow statically in LB medium (Fig. 1b and Figs. S3, S4). To quantitatively follow the aggregation process, we borrowed concepts in polymer phase separation[42] and measured the characteristic length $\xi$ of the system (Fig. 1b, c and Fig. S5). $\xi$ is close to 1 μm (one cell length) in the first few hours, indicating a loosely packed cell population – the Δ*ABC* mutant is known to form expanded structures due to osmotic swelling of VPS[39]. $\xi$ starts to increase sharply at a characteristic time $t_0 = 10$ hours after seeding and reaches a plateau value close to 50 μm after ~17 hours. By tracking the time evolution of $\xi$, we found a robust power law of $\xi \propto t'^\alpha$ with an exponent of $\alpha \approx 0.55$ within the experimentally accessible time window. This exponent is close to that observed during the phase separation of viscoelastic materials[42,43].

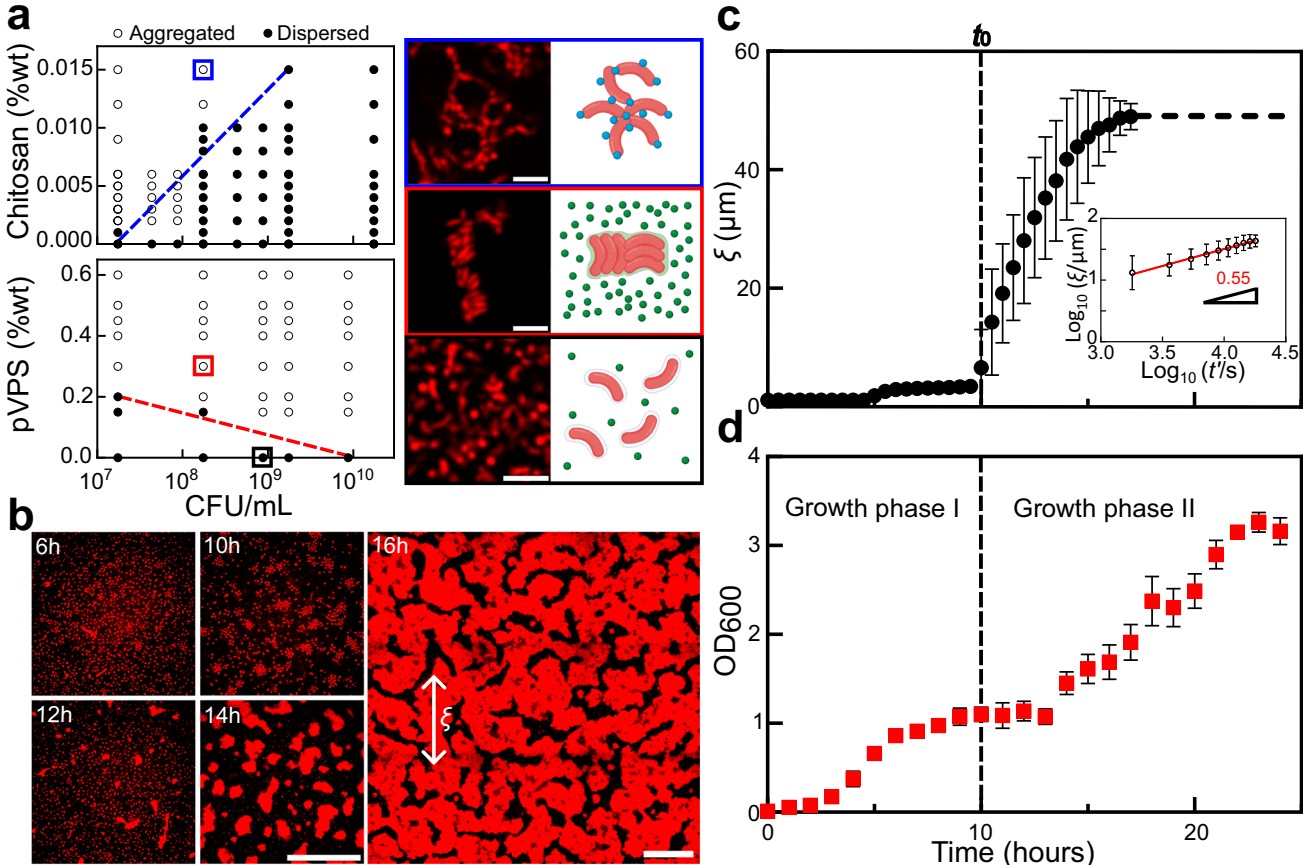

**Fig. 1 | VPS is not attractive to *Vc* cells. a** *Left*: Phase diagrams were generated by mixing non-VPS-producing cells and polymers at the indicated concentrations and visually scoring the cultures as either aggregated (open circles) or dispersed (filled circles) after 6 h. *Right*: Representative images were collected 6 h after mixing non-matrix-producing *Vc* cells and polymers (from *top* to *bottom*: chitosan, purified VPS, and control without any polymer), at $z = 4\,\mu$m above the glass surface. Shown on the right is the schematic for each configuration created with BioRender.com. pVPS stands for purified VPS. Cells constitutively express mRuby3. Scale bars = 5 μm. Colors correspond to the two aggregation mechanisms (blue for bridging-aggregation and red for depletion-aggregation, respectively). %wt = weight percent. **b** Cross-sectional confocal images (at $z = 4\,\mu$m above the glass surface) during the growth of a culture from a VPS-producing strain ($\Delta rbmA\Delta bap1\Delta rbmC$ or $\Delta ABC$ in short). Cell membranes were stained with FM 4-64 (2 μg/mL). $\xi$ indicates the characteristic size of the cell clusters. Scale bars = 50 μm. **c** Evolution of averaged $\xi$ versus time $t$ in cultures of VPS-producing cells (images taken at $z = 4\,\mu$m above the glass surface). Inset: Normalized $\xi$ versus rescaled time $t' = t - t_0$ on log scale. $t_0$ (indicated by the dashed line) corresponds to the onset of the spontaneous aggregation. **d** Growth curve of the $\Delta ABC$ culture. All data are shown as mean ± SD ($n = 4$ biological replicates). Aggregates were disassembled via vortexing before each OD$_{600}$ measurement. Source data are provided as a Source Data file.

Alongside the analysis above, we manually measured the optical density of the cell populations over time under the same growth conditions (see Methods). We observed two distinct growth phases reminiscent of diauxic behavior. The cell culture underwent exponential growth until reaching the first plateau at approximately 8-9 hours, followed by a second exponential growth phase starting around 13 hours (Fig. 1d). Importantly, the onset of depletion-aggregation ($t_0$) lies in the middle of the first plateau preceding the second growth phase. We therefore hypothesized that the onset of spontaneous depletion-aggregation is triggered by a change in the cell-matrix interaction, rather than simply due to an increasing cell count over time.

**Surface remodeling underlies spontaneous depletion-aggregation**

To test this hypothesis, we mixed pVPS with $\Delta ABC$ cells that were taken from different growth phases, fixed with 4% paraformaldehyde, and remapped the 2D phase diagram. For cells harvested in growth phase I (8 h), we observed a significant upward shift of the phase boundary compared to the 5Δ cells (Fig. 2a): at the same bacterial and polymer concentrations, it is harder for depletion-aggregates to form (Fig. S6). In contrast, $\Delta ABC$ cells harvested in growth phase II (20 h) exhibit a

phase diagram similar to the 5Δ cells (Fig. 2b). These results suggest that the biophysical interactions between cells and VPS changes during biofilm growth.

We speculate that in the early stage of biofilm growth (before the first plateau), while VPS molecules are being actively extruded from the biosynthesis machinery[21], they remain anchored to the cell surface such that cells are coated with VPS. Hence, the repulsive cell-VPS interaction should be partially counteracted by the interaction between surface-anchored VPS and the VPS network (Fig. S7). As the nutrient level of the media diminishes, cells cease to make VPS, and hence their surfaces may no longer be covered by VPS. This surface remodeling process is a likely candidate for triggering the depletion-aggregation. As an indirect test of this hypothesis, we treated $\Delta ABC$ cells harvested in growth phase I with RbmB, the native polysaccharide lyase[25,32], to trim VPS off of the cell surface. Indeed, thus-treated cells exhibit a phase diagram closer to those from 5Δ cells or $\Delta ABC$ cells fixed in growth phase II (Fig. 2c), than to that of the untreated cells. This is consistent with our hypothesis that surface remodeling unmasks the repulsive cell-VPS interaction and thereby triggers the onset of depletion-aggregation.

In addition to the biochemical approach described above, we performed another test of the same hypothesis by manipulating the

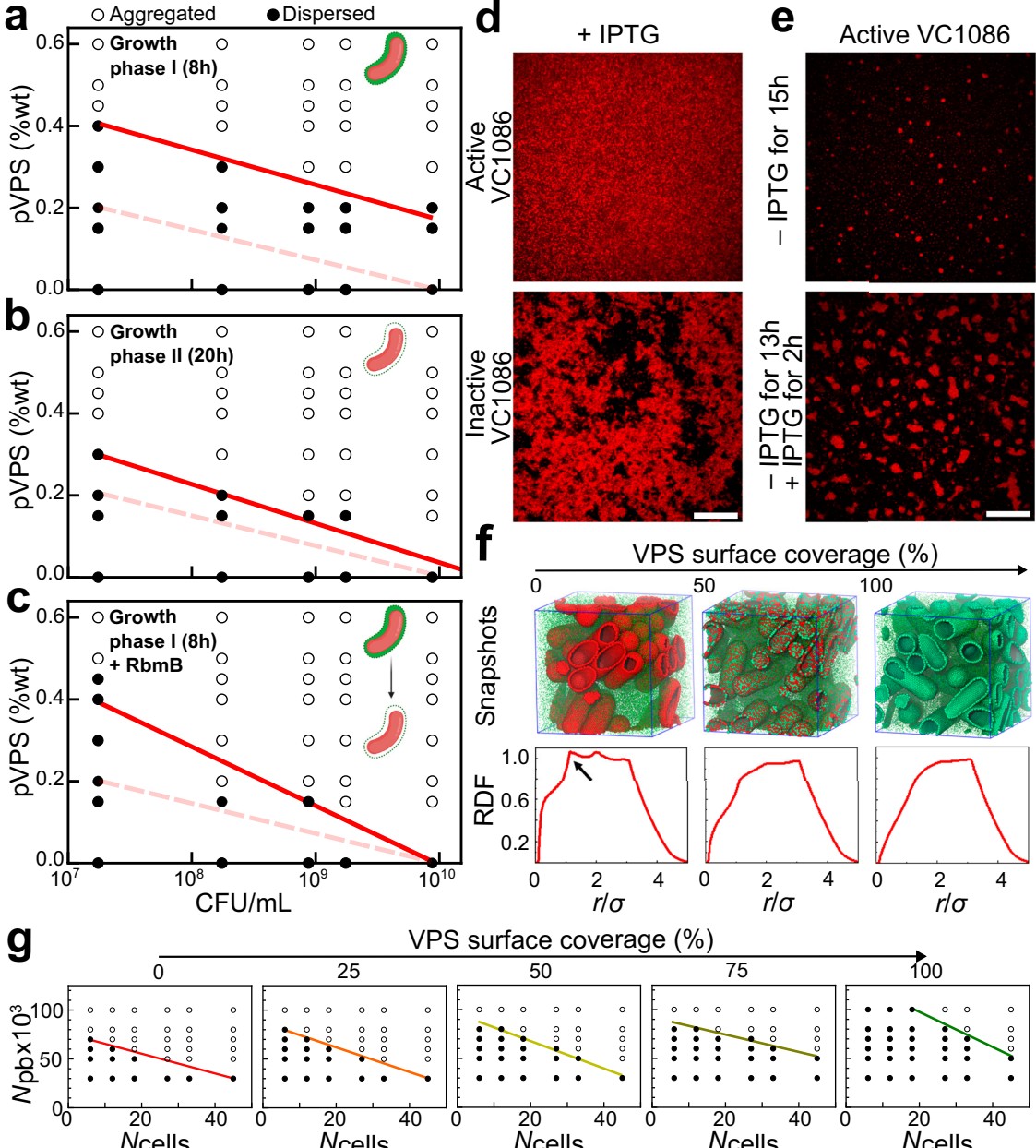

**Fig. 2 | Surface remodeling underlies the spontaneous depletion-aggregation in VPS-producing culture. a–c** Phase diagrams were generated by mixing pVPS and VPS-producing cells (ΔABC cells) chemically fixed using 4% paraformaldehyde at (**a**) growth phase I (8 h), (**b**) growth phase II (20 h), and (**c**) growth phase I followed by incubation overnight with RbmB (0.05 mg/mL), at the indicated concentrations and visually scoring the cultures as either dispersed (filled circles) or aggregated (open circles). Red solid lines correspond to the phase boundary and dashed lines correspond to the original phase boundary obtained with 5Δ cells. Shown as an inset in each phase diagram is a schematic of the cell configuration for each cell type, respectively, created using BioRender.com. **d, e** Controllable transition of depletion-aggregation (images shown at z = 4 μm above the glass surface). Representative images were shown from four repeated experiments. **d** Cross-sectional confocal images of a strain carrying a plasmid with IPTG inducible expression of VC1086, a c-di-GMP phosphodiesterase (*top*) and an isogenic strain carrying a control plasmid with an inactive version of VC1086 (*bottom*). Both strains were grown in the presence of 1 μM IPTG overnight and visualized. Scale bar = 10 μm. **e** The strain carrying active VC1086 was grown for 15 h without IPTG (*top*) or

without IPTG for 13 h followed by IPTG treatment for 2 h. Cell membranes were stained with FM 4-64. Scale bar = 10 μm. **f** Coarse-grained simulation of the depletion-aggregation phenomenon controlled by surface remodeling. *Top*: Snapshots of cellular arrangements following 100 million simulation steps, at varying VPS surface coverages (from left to right, 0%, 50%, and 100%). Individual VPS polymers are shown as green spheres. Patches on bacterial cell surfaces are colored depending on whether they are covered with VPS (green) or not (red). *Bottom*: The corresponding radial distribution function (RDF). Arrow indicates an emerging peak characteristic of depletion-induced parallel arrangement of the rod-shaped cells. **g** Simulated phase diagrams as a function of VPS surface coverage scored as either dispersed (filled circles) or aggregated (open circles). The ranges considered for the number of cells, number of polymer beads (pb), and VPS surface coverage are from 6 to 45 cells, 30,000 to 100,000 beads, and 0 to 100% VPS surface coverage, respectively, in a simulation box of 6.2 × 6.2 × 6.2 μm³. Solid color line from red (0% VPS coverage) to green (100% VPS surface coverage) in each phase diagram shows the boundary between the aggregated and dispersed phases. Source data are provided as a Source Data file.

intracellular c-di-GMP level. Our rationale is based on the well-established positive correlation between intracellular [c-di-GMP] and VPS production[16,17]. We introduced into the $\Delta ABC$ strain a plasmid encoding VC1086, a phosphodiesterase that degrades c-di-GMP, and produced another isogenic strain with a plasmid containing an inactive version of VC1086 as negative control[44]; both are inducible by iso-propyl β-D-1-thiogalactopyranoside (IPTG). It was shown previously that the induction of the active version of VC1086 can significantly reduce the intracellular [c-di-GMP][44], which should abolish VPS pro-duction. Indeed, we first showed that inducing the active version of VC1086 throughout growth leads to a culture of freely swimming cells, consistent with the abolishment of VPS production (Fig. 2d, top and Fig. S8). In contrast, the control strain harboring an inactive version of VC1086 forms depletion-aggregates (Fig. 2d, bottom) as the parental strain. To test if we can artificially induce the formation of depletion-aggregates by enforcing a physiological change, we grew the strain with the active VC1086 without IPTG for 13 hours to accumulate suf-ficient VPS, followed by a 2 h exposure to IPTG (Fig. 2e, bottom). During the IPTG exposure, VPS production should be turned off so that cell surfaces are no longer coated with VPS. Consistent with the surface remodeling idea, large aggregates were visible at the end of this experiment, whereas the control sample that grew for the same duration without IPTG had only started to form small nucleates of aggregates (Fig. 2e, top); the control plasmid with the inactive VC1086 does not cause this difference (Fig. S8). In this set of experiments, the IPTG-induced decrease in the c-di-GMP level and the resulting cessation of VPS production trigger a premature onset of cell aggregation due to a rapid shift in the cell-VPS interaction. During unperturbed biofilm growth, the surface remodeling process may be more gradual.

## Molecular dynamics simulation recapitulates the phase transition

To quantify the relationship between VPS production, surface remo-deling, and aggregation, we conducted molecular dynamics (MD) simulations[45]. VPS polymers are modeled as freely diffusing beads; cells are modeled as spherocylinders, whose surfaces are modeled as a triangular network of 1179 nodes with the same size as the polymer spheres. The interaction between the VPS beads and the cellular sur-face nodes depends on the identity of the nodes: nodes that represent the native cell surface are implemented with a repulsive interaction with the polymer beads (1 $k_BT$), whereas nodes that represent VPS-coated surface were modeled with a much weaker repulsion to the beads (0.1 $k_BT$). Figure 2f shows the results of implementing 50,000 polymer beads and 33 cells with 0%, 50%, and 100% VPS coverage, in a simulation box of $6.2 \times 6.2 \times 6.2$ μm³. With 0% VPS surface coverage, depletion occurs readily (Fig. S7). The characteristic peak in the radial distribution function (RDF) at 1.14 μm (Fig. 2f, left) implies that cells align in parallel with close surface-to-surface contact (Fig. S7b). As the VPS surface coverage increases, the peak disappears and cells no longer align in parallel (Fig. 2f, right). Simulated phase diagrams as a function of VPS surface coverage clearly show that as the VPS surface coverage decreases (Fig. 2g and Fig. S7), the phase boundary shifted downwards – the system becomes easier to phase separate via depletion. In the experiment, we suggest that nutrient limitation induces the decrease in VPS surface coverage on the cells, which in turn increases the repulsion between VPS and cells and consequently triggers depletion-aggregation.

## Crosslinking leads to bridging-aggregation

Many EPS types are crosslinked by accessory proteins to provide fur-ther mechanical stability to the biofilm[6,46,47]. In *Vc*, it has been sug-gested that VPS crosslinking can be achieved by Bap1 and RbmC, which share a conserved β-propeller domain that binds to VPS[25,28,30]. To confirm this, we first showed that Bap1 caused the formation of large VPS clumps when the two were mixed together using fluorescently

labeled wheat germ agglutinin (WGA)[21] (Fig. 3a); similar results were obtained with negative-staining electron microscopy (Fig. S10). In these clumps, Bap1 and VPS signals colocalize, consistent with prior observation in biofilms[21] and with the in vitro binding assay of Bap1 with VPS[30].

Next, we investigate how VPS crosslinking affects the cell aggre-gation process. When we added Bap1 to a growing $\Delta ABC$ culture, we found disordered cell aggregates loosely connected by the VPS-Bap1 network (Fig. 3b and Fig. S11). Our interpretation of the result is that cells with surfaces coated with VPS are able to anchor to the cross-linked VPS/Bap1 network, through the same crosslinking function of Bap1. Taking this a step further, we wondered if Bap1 can also form direct cell-cell junctions via binding two VPS-coated cells simulta-neously. To test this possibility, we mixed Bap1 with fixed, VPS-coated cells (i.e. $\Delta ABC$ cells in growth phase I) but without exogenously added VPS. As we hypothesized, this experiment produced cellular aggre-gates with a loose, disorganized morphology characteristic of bridging-aggregates (Fig. 3c). Single-cell analysis of these aggregates shows that cell orientation, unlike the case of depletion-aggregates, is uncorrelated between neighboring cells (Fig. 3d). Moreover, the cell-to-cell distance is significantly larger (peak at 2.35 μm) in the bridging case than in the depletion case (peak at 0.95 μm), due to the combined effect of random cell orientation and the existence of polymer between the cells. When we labeled Bap1 with GFP and stained the aggregates with fluorescent WGA, we found that cell surfaces were covered with both VPS and Bap1 (Fig. 3d); this serves as direct evidence for the existence of surface-associated VPS on cells during the early biofilm growth phase. Finally, the slope of the 2D phase boundary became positive (Fig. 3c), similar to the cases for PLL and chitosan, again supporting that the observed aggregates in this case arise from bridging.

Synthesizing the above observations, we propose a conceptual model in which Bap1 can either directly bridge cells whose surfaces are coated with VPS molecules due to active secretion, or enable cells to anchor to an existing VPS network (Fig. 3e). We also reproduced such bridging-aggregation in MD simulations where we assigned attractive interactions between the polymer beads and cell surfaces (Fig. S7d). We have also varied the attraction strengths in the simulation (Fig. S7e) and showed that a strong attraction (50 $k_BT$) gives cluster morpholo-gies similar to the experimental data, in which cells are randomly oriented. This potentially suggests that the cell-cell interaction con-ferred by VPS crosslinking in the experiment is also strong compared to thermal energy; indeed, we do not see much thermal fluctuation of cells in the biofilm. Depending on the concentration of polymer beads in the MD simulation, cells can be either directly bridged by one layer of beads or a clump of beads, again consistent with the two scenarios observed in the experiment (Fig. S7).

## Regulation of VPS is the key to inversion of cell-matrix interaction

To study the combined effect of VPS crosslinking and surface remo-deling on biofilms, we grew a $\Delta rbmA$ strain that contains Bap1 and RbmC in the biofilm matrix. In biofilms formed by this strain (but not strains missing Bap1/RbmC), VPS can be labeled in situ with fluorescent WGA[21,48], which allowed us to directly quantify VPS production during biofilm development (Fig. 4a). We found that in the first growth phase, the biofilm consisted mainly of cells encased by VPS ( > 85%) (Fig. 4b). In the second growth phase, this proportion declined precipitously, down to zero at 20 h. Moreover, cells that are not encased by VPS formed aggregates that are morphologically similar to the aggregates we attributed to depletion (Fig. S12a).

To complement the in situ VPS-staining experiment, we quantified the expression level of the *ups*-II operon using the *upsL-lux* reporter[20]. We observed that cells exhibit an early surge in *upsL* expression, fol-lowed by a drastic reduction when entering the second growth phase,

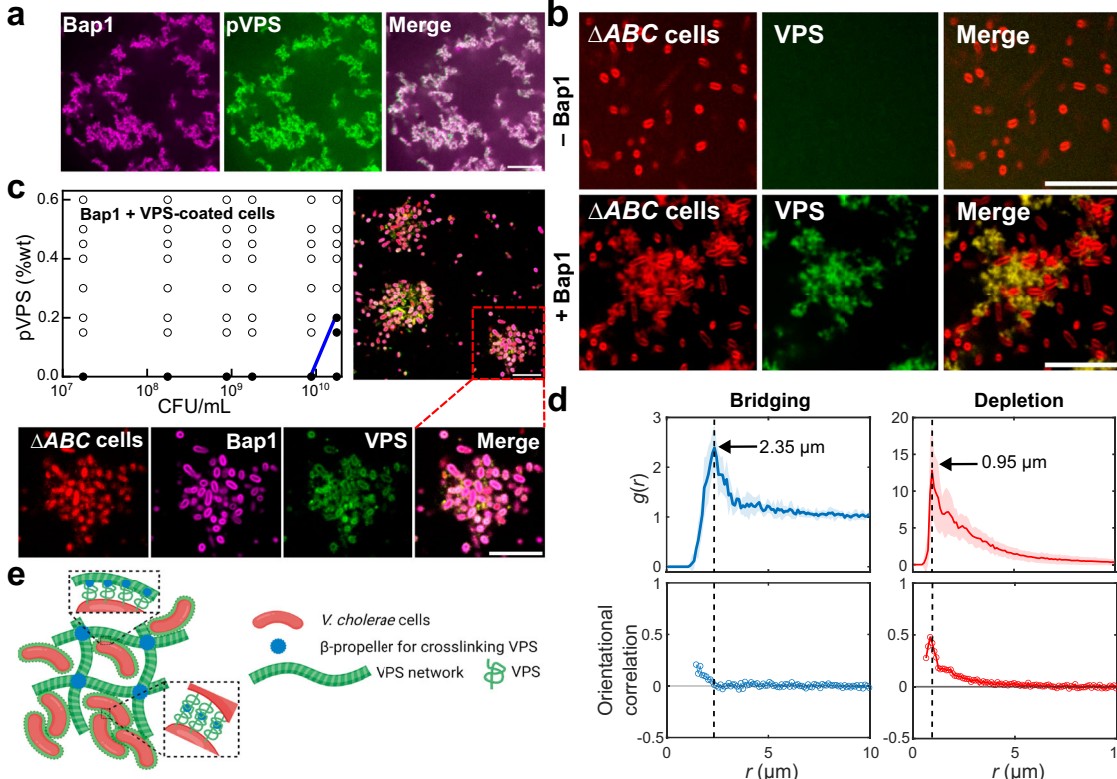

**Fig. 3 | Crosslinking of VPS by Bap1 leads to bridging-aggregation. a** Cross-sectional images from a mixture of Bap1-GFP at 1.2 mg/mL (pseudocolored in magenta) and pVPS at 0.6 mg/mL stained with wheat germ agglutinin (WGA) conjugated to AlexaFluor647 (pseudocolored in green). Representative images were shown from four repeated experiments. **b** Cross-sectional images of ΔABC culture after 8 hours of growth, with or without 1 mg/mL Bap1 (images taken at z = 4 μm above the glass surface). Cell membranes were stained with FM 4-64 (red). Representative images were shown from four repeated experiments. **c** *Upper left*: Phase diagram generated by mixing Bap1 with ΔABC cells constitutively expressing mRuby3 fixed in growth phase I, at the indicated concentrations and visually scoring the cultures as either dispersed (filled circles) or aggregated (open circles). Blue solid line corresponds to the phase boundary. *Upper right*: Cross-sectional images showing aggregates of ΔABC cells bridged by exogenously added Bap1 (at z = 9.35 μm above the glass surface). *Bottom*: Zoomed-in images showing the localization of ΔABC cells constitutively expressing mRuby3 (red); GFP-labeled Bap1

(pseudocolored in magenta); VPS polymers stained with WGA-647 (pseudocolored in green); and overlaid image showing the colocalization of Bap1 and VPS surrounding each cell. Scale bars = 10 μm. **d** *Top*: Radial distribution function g(r) of cell centroids, for the bridging case corresponding to experiment in (**c**) (*left*) and the depletion case corresponding to 5Δ cells plus purified VPS (*right*). Data are presented as mean values from measurements taken from distinct samples ± SD (shade around the solid line). *Bottom*: Orientational correlation as a function of distance between the cell centroids, r. Orientational correlation is defined as $\langle\frac{1}{2}(3\cos^2\theta_{ij} - 1)\rangle_r$, in which $\theta_{ij}$ corresponds to the angle between the orientations of cells i and j and the angle bracket denotes the average over all cell pairs at a cell-to-cell distance r. Cell centroids and orientations were obtained by single-cell segmentation. Data are presented as mean values from measurements taken from distinct samples. **e** Schematic of how Vc cells form biofilms via cell-VPS and cell-cell interaction mediated by crosslinking, created using BioRender.com. Source data are provided as a Source Data file.

turning off *upsL* expression completely by 15 hours (Fig. 4c). The downregulation of matrix production is consistent with the decrease in VPS staining in the second growth phase (Fig. 4b, c).

We interpret these results as corresponding closely with results from the aggregation assays, as follows. In early biofilm growth, cell surfaces are covered with VPS and therefore, each biofilm is essentially an aggregate of cells bridged by the crosslinked VPS network and/or directly by Bap1 (and RbmC). As nutrient limitation sets in, the disappearance of VPS on cell surface inverts the cell-matrix interaction from attractive to repulsive and converts the system from bridging- to depletion-aggregation.

We recapitulated such transition from bridging- to depletion-aggregation in MD simulations. Specifically, we modeled the inversion of the cell-matrix interactions by starting from 100% attractive polymer beads and gradually increasing the fraction of polymer beads that are repulsive to each other and to the cell surfaces (Fig. 4d). Interestingly, we found that only a small fraction of attractive beads (1%) is needed to change the system from depletion (characterized by a sharp peak at 1.14 μm) to aggregation (characterized by a broad peak at a higher cell-to-cell distance). Hence, we expect the bridging mechanism to dominate in biofilms until the cell surfaces are devoid of VPS.

## RbmB facilitates the inversion of cell-matrix interaction

While surface remodeling can be achieved via downregulation of VPS, the results shown in Fig. 2c suggest that this process can be accelerated by the enzymatic activity of RbmB. Indeed, adding RbmB to VPS-coated cells abolishes the Bap1-mediated bridging-aggregation, presumably by trimming VPS off from the cell surfaces (Fig. S9d). On the other hand, RbmB has been implicated to be required for dispersal from prior genetics studies[14,17,25], although the exact mechanism remains unclear.

To put these results into the biofilm context, we added purified RbmB during the early development of a ΔrbmA biofilm (6 hours). We observed the immediate disappearance of the VPS signal on the cell surface (< 30 mins) and accelerated depletion-aggregation (Fig. 5a, top). A similar effect can be obtained by inducing *rbmB* expression from a plasmid (Fig. 5a, *middle*). These results show that RbmB can facilitate dispersal in accordance with previous reports[14,25,31], and further suggest that it may do so mainly by trimming off surface-anchored VPS, which changes the cell-matrix interaction.

As an alternative way to demonstrate the role of RbmB, we further deleted *rbmB* in Vc and visualized the spatial distribution of the cells and VPS. Indeed, the surface VPS signal remains

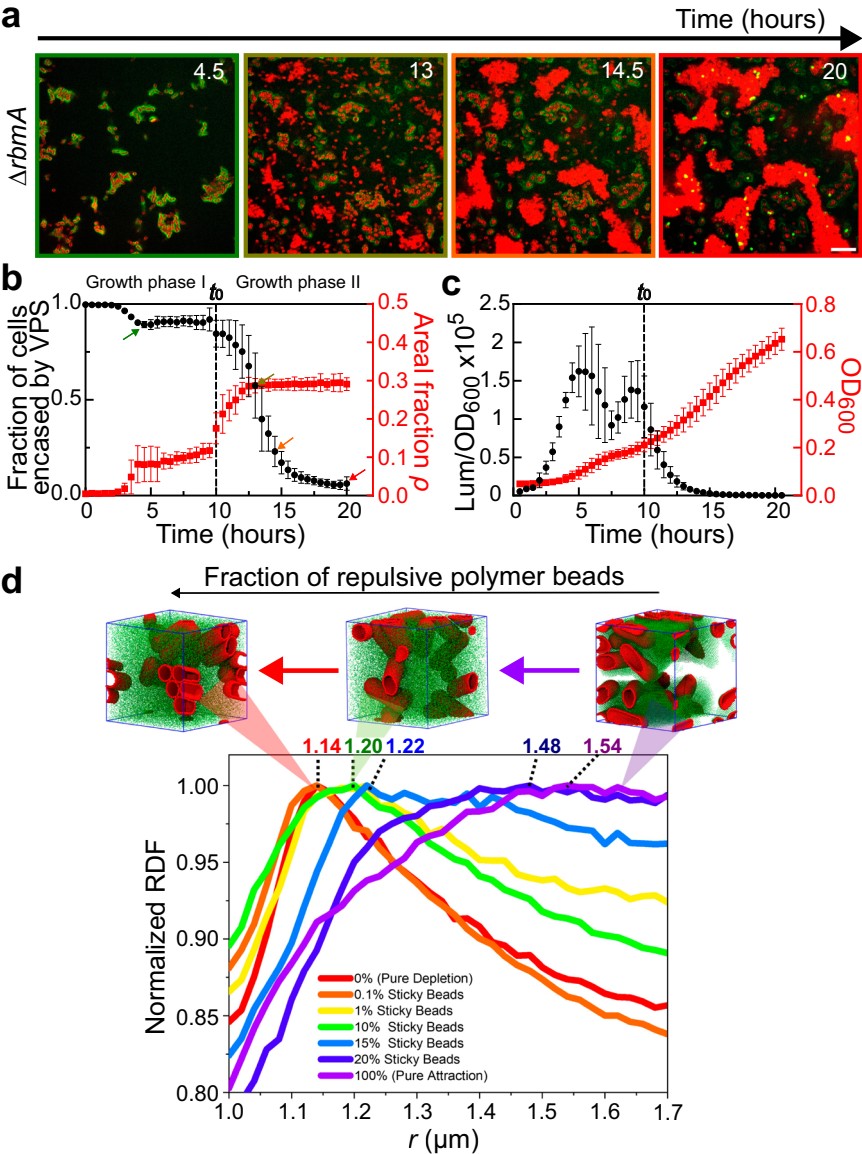

**Fig. 4 | Quantification of VPS production during biofilm growth. a** Time-course of growth from 4.5 to 20 hours of a biofilm formed by the $\Delta rbmA$ strain (images shown at $z = 4\,\mu m$ above the glass surface). Cell membranes were stained with FM 4-64 (red) and VPS was stained with WGA conjugated to Oregon Green (green). Note at the late stage, some dead cells with exposed cell walls were also stained with WGA. Scale bars = 10 μm. **b** Quantification of the fraction of cells encased by VPS (black circle) and the total cell density approximated as the areal fraction $\rho$ in the imaging plane (red square) during 20 hours of the growth of $\Delta rbmA$ biofilms. The color of the image border in (**a**) corresponds to the time points highlighted by arrows of the corresponding colors in (**b**). Data are presented as mean values $\pm$ SD ($n = 5$ biological replicates). **c** Quantification of *upsL* gene expression in $\Delta rbmA$ biofilms through the measurement of luminescence from the pBBRlux-*upsL*

reporter, normalized by $OD_{600}$. Data are presented as mean $\pm$ SD ($n = 4$ biological replicates). **d** Simulation results with a variable fraction of polymer beads that are repulsive to each other and to the cell surface. The simulation box ($6.2 \times 6.2 \times 6.2$ μm³) contains 80,000 polymer beads and 12 cells. When the polymer beads are all attractive to each other and to the cell surface (magenta), a broad peak with a peak position away from close contacts is observed, consistent with the experimental RDF in the bridging case. As the fraction of repulsive polymer beads increases, the peak becomes sharper and the peak position shifts to shorter cell-cell distances, and eventually reduces to the pure depletion case (red). Shown on top of the curves are corresponding simulation snapshots, in which cells are shown in red and polymer beads in green. See Supplementary Table S2 for simulation parameters. Source data are provided as a Source Data file.

stable over time, consistent with the absence of VPS cleavage. Interestingly, no depletion-aggregates were observed in the $\Delta rbmA\Delta rbmB$ biofilm at the late stage (Fig. 5a, bottom and Fig. S12b): new cells that escaped the VPS envelope grew as individual planktonic cells, indicating the absence of free VPS polymers in the solution. We conjectured that in the $\Delta rbmA\Delta rbmB$ mutant the surface-anchored VPS cannot be released, causing the absence of free VPS polymers in the solution; cells can only disperse via downregulation of VPS in this case.

**Cells are included in biofilms only when actively producing VPS**
Integrating all the information presented above, we propose a multi-step process that underlies the dynamics of polymer-driven bacterial community formation and disintegration (Fig. 5b). Crucially, the recognition of community membership is closely tied to the active secretion of VPS: cells coated by VPS – which only occurs when the exopolysaccharide is actively being produced – are recognized by the existing VPS-Bap1 network or other VPS-coated cells and consequently included within the biofilm through bridging-aggregation. Conversely,

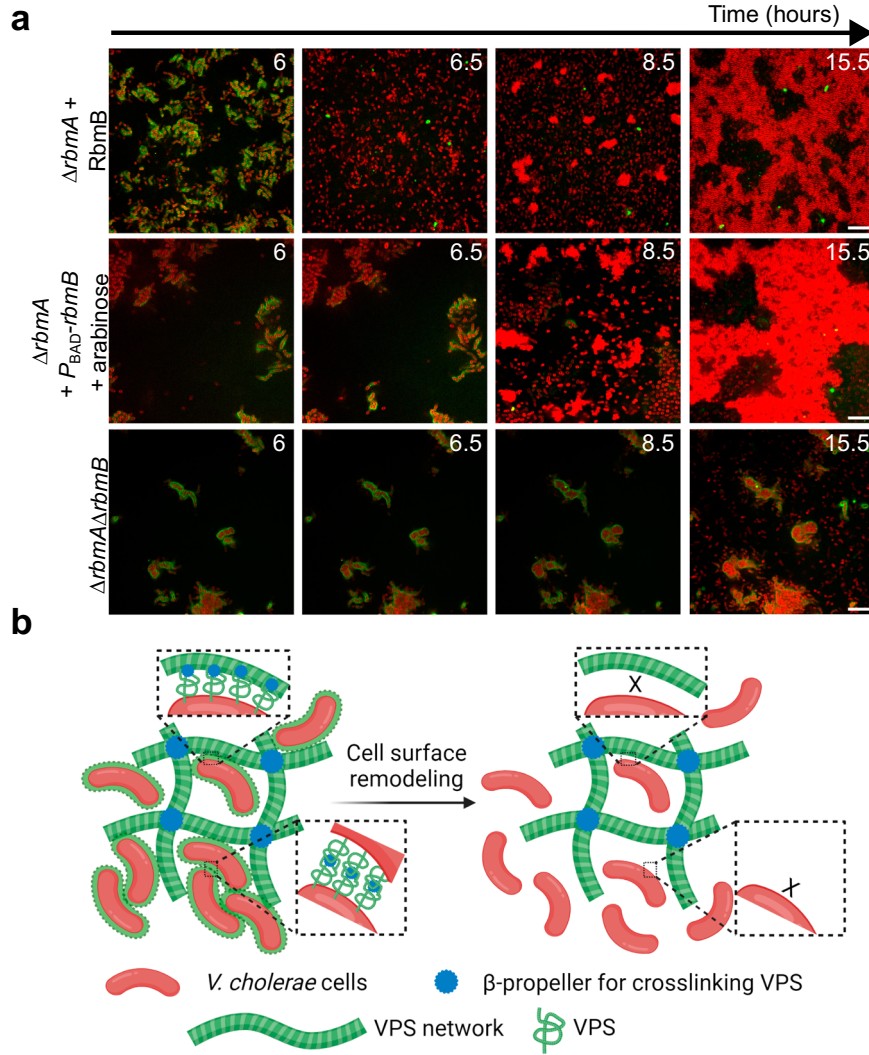

**Fig. 5 | Inversion of cell-matrix interaction controls biofilm disassembly.**
**a** Cross-sectional images at $z = 4$ μm above the glass surface from a time course of biofilm growth of $\Delta rbmA$ (*top*), $\Delta rbmA + P_{BAD}$-$rbmB$ (*middle*), and $\Delta rbmA\Delta rbmB$ (*bottom*). After 6 hours of biofilm development, RbmB (0.05 mg/mL) or arabinose (0.05 %wt) was added for the top and middle rows, respectively. Cell membranes were stained with FM 4-64 (red) and VPS was stained with WGA-Oregon Green (green). Scale bars = 10 μm. **b** Schematic of the interplay between *Vc* cells and VPS network during biofilm development, created using BioRender.com. See text for more details.

cells devoid of the VPS layer are excluded from the biofilm; these cells form expelled depletion-aggregates when free VPS accumulates in the culture. This pattern of interaction between VPS-producing and non-producing cells follows the patterns of so-called "greenbeard" traits[49,50] that confer cooperative fitness benefits of some kind (in this case, biofilm formation) and, at the same time, provide a mechanism for selectively benefiting other cells that are also making the matrix.

### Inversion of cell-matrix interactions facilitates dispersal

We further propose that the surface remodeling process and resulting inversion of cell-matrix interactions are crucial for the disassembly of the biofilm community and dispersal of constituent cells upon nutrient limitation. To test this ecological idea, we challenged the $\Delta rbmA$ biofilms at various time points with a washing step that mimics flows likely encountered by many microbial communities in nature[51] (Fig. 6). We chose a $\Delta rbmA$ strain in this set of experiments because it can sustain washing through the adhesive function of Bap1 and RbmC[25,26,30,48]. We choose two time points for introducing a pulse of flow: 12 hours, shortly after the onset of depletion-aggregation, and 17 hours (Fig. 6a), by which time large depletion-aggregates have already formed. In the former case, approximately 75% of the biomass persists, while in the

latter this fraction reduces to 50% (Fig. 6b). This suggests that the depletion-aggregates exhibit an increased susceptibility to removal. We further quantified the fraction of cells present after washing, separating cells encased by VPS from those that are not (Fig. 6c). Cells encased by VPS largely remain in place, whereas non-VPS-encased cells in the depletion-aggregates are removed from the system, presumably because they are no longer protected from shear stress by the matrix. Once transported to pristine surfaces in fresh LB medium, these depletion-aggregated $\Delta rbmA$ cells form biofilm structures indistinguishable from the original biofilm (Fig. 6f) – they behave as dispersed cells.

Our results suggest a dual role of VPS: 1) it acts as the main structural element during biofilm growth; 2) it excludes non-VPS producing cells and pushes them into depletion-aggregates that are easily removed by flow. The latter function can be leveraged during biofilm dispersal by ensuring that the dispersing cells no longer adhere to the biofilm and thus can be transported to a new environment where there are potentially new food resources (Fig. 6g). Note that in this picture, matrix degradation is not necessarily required for cell dispersal (as sometimes assumed in earlier reports[14,15]); surface-trimming, either by the enzymatic activity of RbmB or through downregulation of

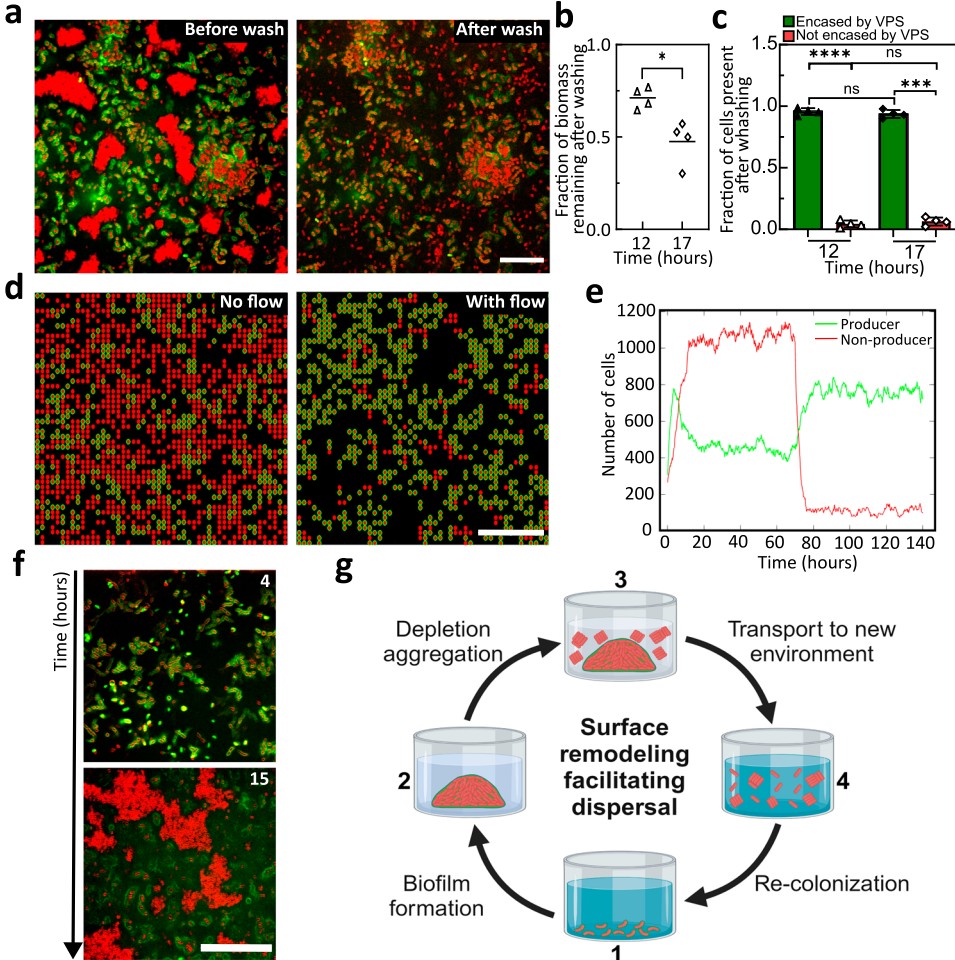

**Fig. 6 | Surface remodeling and aggregation facilitate dispersal. a** Cross-sectional confocal images at *z* = 6 μm above the glass surface before (*left*) and after (*right*) a washing step performed after 17 hours of growth for a Δ*rbmA* biofilm. Cell membranes were stained with FM 4-64 (red) and VPS was stained with WGA-Oregon Green (green). Scale bars = 10 μm. **b** Quantification of the fraction of biomass remaining after the washing step after 12 or 17 hours of growth for a Δ*rbmA* biofilm (*n* = 4 biological replicates). Statistical analysis was performed using unpaired, two-sided *t*-test with Welch's correction to differentiate the fraction of biomass remaining after the wash for 12- and 17-hour-old biofilms. *p* = 0.0203. **c** Quantification of the fraction of *Vc* cells encased (green) or not encased (red) by VPS after the washing step for 12- and 17-hour-old biofilms, respectively. Statistical analysis was performed using an unpaired, two-sided *t*-test with Welch's correction to differentiate the same cellular population between experiments at 12 and 17 hours, whereas a paired *t*-test was used to differentiate

each cellular population at the same time point. ns stands for not significant; ***\*p* < 0.001; \*\*\*\**p* < 0.0001. Exact *p* value from left to right: <0.0001, 0.4041, 0.4041, 0.0001. Data are presented as mean values from measurements taken from distinct samples ± SD (*n* = 4 biological replicates). **d** Representative snapshots of results from a spatial stochastic model on a 2D grid including producer (P in red with green outline) and non-producer (nP in red) cells, before (*left*) and after (*right*) turning on flow. Scale bars = 10 μm. **e** Quantification of the number of P and nP cells before and after turning on flow (at *t* = 70 hours). **f** Representative confocal images of a Δ*rbmA* biofilm seeded by the depleted-aggregated cells harvested in the washing step in (a) and grown in fresh medium (at *z* = 6 μm above the glass surface). Cell membranes were stained with FM 4-64 (red) and VPS was stained with WGA-Oregon Green (green). Scale bar = 50 μm. **g** Schematic of the *Vc* biofilm lifecycle corresponding to the experiments in (**a**–**f**), created using BioRender.com. Source data are provided as a Source Data file.

VPS, is sufficient to change the cell-matrix interaction to a repulsive one, allowing cells to leave the existing biofilm.

To generalize our observations and generate more mechanistic insights, we employed a dynamical systems model containing both producer (P) and non-producer (nP) cells. Our lattice-based stochastic model includes the processes of cell division, removal, and a density-dependent interconversion between P and nP cells: higher population densities favor the conversion from P to nP cells and penalize the reverse conversion (see Supplemental Methods). Importantly, P cells have a lower removal rate in flow than nP cells due to surface adhesion, and this difference is amplified if a P cell is surrounded by other P cells. In a steady state, we found that the system is characterized by an overabundance of nP cells without flow (Fig. 6d, left), due to a higher growth rate and dominating flux from P to nP at high cell density. Once the flow starts (Fig. 6d, right), nP cells are rapidly washed away while P

cells benefit from a reduced wash-out rate resulting from their adhesion to the surfaces and neighboring P cells, creating stable clusters (Fig. 6d, e). Also, the removal of the nP cells reduces the overall density, reversing the conversion flux from nP to P cells. Our simple model not only recapitulates key experimental observations in Fig. 6a–c but also enables straightforward extension to other ecological scenarios (Fig. S13).

## Discussion

To summarize, using the model biofilm-former *Vibrio cholerae*, we showed that the cell-matrix interaction is not always attractive in biofilms but instead adapts to different stages during biofilm development. Specifically, only when *Vc* cells are actively secreting VPS do they adhere to each other through Bap1, which crosslinks VPS and therefore acts as a bridge between cells and between the cell and the matrix. This

picture explains many perplexing behaviors of *Vc* biofilms: biofilm-forming cells naturally segregate from and repel non-biofilm-forming mutants[9,10,39] because VPS acts not only as a structural element but also as a recognition mechanism. In other words, the *vps* genes along with genes encoding matrix proteins behave as a "greenbeard" gene complex that simultaneously confers a benefit at the group level (such as resistance to shear stress, predation, and chemical attack) and also ensures that only other cells investing in matrix production receive its benefits[7,49,50]. Biochemically, the intrinsic cell-VPS repulsion may come from the electrostatic repulsion between the negatively-charged cell surface and the carboxylate groups on C6 of the multiply modified L-gulose moiety in VPS[19]. Although we have focused on Bap1 here, we expect that the role of RbmC is similar due to the high structural and functional similarity between their β-propellers that bind to VPS[28,30].

As biofilms grow and nutrient availability diminishes, cells cease to express and coat their surfaces with VPS. This results in an inversion of cell-matrix interactions, which resolves another dilemma: the VPS matrix both provides structural integrity to biofilms when the nutrients are abundant and also facilitates, not inhibits, cell dispersal through the intrinsic cell-VPS repulsion when the nutrients are scarce. Therefore, complete degradation of the matrix is not necessary for dispersal because cells can efficiently leave the scaffold once their surfaces are devoid of VPS. More broadly, our results highlight the critical need to identify universal biophysical mechanisms underlying biofilm development[52] as a foundation to discover new strategies to disrupt biofilms in disease and industrial contexts.

One major drawback of the current study is the exclusion of RbmA, the accessory protein responsible for biofilm compactness and cell-cell cohesion[22,24]. RbmA has been shown to bind to VPS[22]; the crystal structure of RbmA suggests a potential binding site to cell surfaces[27], and immunostaining of RbmA shows that it coats the surface of biofilm-dwelling cells[21,26]. However, RbmA alone does not bind to cell surfaces in the absence of VPS. Hence, we propose that one potential function of RbmA is to provide another link between the matrix and *Vc* cell surface to work against the inherent cell-VPS repulsion. During dispersal, RbmA is degraded by HapA/IvaP/PrtV[53], potentially reestablishing the repulsive cell-VPS interaction to enable dispersal. While we used a high c-di-GMP background to ensure robust biofilm formation, we expect the nature of the cell-VPS interaction to remain the same in wild-type *Vc*. Indeed, we were able to reproduce some of the results in the wild-type background in a confined geometry that enhances the local concentration of VPS (Fig. S14).

One area for future research is a thorough characterization of the biochemical and biophysical properties of VPS as well as its interactions with other matrix components. Although the chemical structure of VPS was recently elucidated[19,32], other important polymer properties, including molecular weight, persistent length, intra- and intermolecular interactions, are all unknown. Even the biogenesis machinery for VPS is poorly understood, leaving key questions such as how VPS is synthesized and secreted unanswered[54]. This lack of knowledge hampers our understanding of the VPS network and consequently the biofilm structure in *Vc*. Moreover, VPS molecules may also have a different configuration when anchored on the surface – future cryoEM studies may reveal further structural details, as has been done recently for other surface appendages in biofilms[47]. The interaction of VPS with other accessory matrix proteins has been the focus of intensive research recently[22,30], which will likely generate new insights into the *Vc* biofilm matrix.

Even less understood is how VPS is degraded. For example, it remains to be shown where RbmB is localized in the biofilm (extracellular or intracellular). Recent work has started to shed light on the lyase activity of RbmB[32]; putting this biochemical information into the biofilm context in the future will likely solve the mystery of the *Vc* dispersal process. One major question is if the VPS network is even degraded during the dispersal process: reports have shown that *Vc* cells can exit a biofilm in a short time[13,31], leaving behind a "skeleton"[15], which presumably is the intact or partially degraded VPS network. Indeed, our results suggest that as long as the cell surfaces are devoid of VPS and cell-matrix interactions are inverted, dispersing cells will be disconnected and repelled by the network.

In addition to the experimental findings, our work also highlights how computer simulations and modeling can provide quantitative, biophysical insights into bacterial community formation. In the current work, we ignored the complex molecular details of entanglement, self-association, and crosslinking of VPS and treated VPS molecules as spheres with either attractive or repulsive interactions with each other and/or with the cell surface. This simplification allows us to focus on the effect of the inversion of cell-matrix interactions on the biofilm structure. Future development should incorporate physiochemical details of VPS such as its persistent length, molecular weight, and crosslinking to generate realistic, multiscale models for simulating biofilms.

The methodology of using phase diagrams and aggregate morphology to identify the nature of cell-polymer interactions was introduced to the biofilm field by Secor et al.[33]. More generally, depletion has been commonly used to form cellular aggregates, often using nonnative polymers[33,35,36] but also in a few cases, native exopolysaccharides[34]. Depletion itself is a well-studied phenomenon in colloidal science: first described by Asakura and Oosawa in 1954[55], it finds wide use in destabilizing colloidal solutions[35], forming colloidal gels[56], and more recently, inducing the controlled assembly of anisotropic colloids[57]. Polymer bridging is extensively studied as well, particularly in the context of polymer nanoparticle composites[58]. We envision that the incorporation of tools and concepts from colloidal and polymer sciences to biofilm research, as we did here, will continue to yield new perspectives and discoveries in the biofilm field.

Finally, we speculate that a similar phenomenon may be found in other biofilm-formers beyond *Vc*, particularly in species that rely on polysaccharides as the main matrix component[6], although the specific biochemistry of the exopolysaccharides and matrix proteins may likely lead to other scenarios. As a proof of concept, we showed that another model biofilm-former, *Pseudomonas aeruginosa* (*Pa*), can also spontaneously form depletion-aggregates during static growth (Fig. S15). Such depletion aggregation is only observed in a mucoid strain (PAO1Δ*mucA*) that produces high amounts of alginate as the major exopolysaccharide[46,59], and correspondingly, a phase diagram with a negative phase boundary slope was found when mixing non-matrix-producing *Pa* cells with alginate. Further research will be needed to test the generality of our model and perhaps more interestingly, what biochemistry leads to the failure of the biophysical model when it does not apply.

## Methods

Method for dynamical systems modeling is further described in Supplementary Methods.

### Strains, media and materials

All *V. cholerae* strains used in this study are derivatives of the wild-type *Vibrio cholerae* O1 biovar El Tor strain C6706 and are listed in Table S1. All strains harbor a missense mutation in the *vpvC* gene (*vpvC*[W240R]) that elevates the intracellular cyclic diguanylate (c-di-GMP) level[40]. Additional mutations were genetically engineered using natural transformation unless indicated otherwise[60]. Plasmids were transferred into *V. cholerae* strains through conjugation. The *P. aeruginosa* strains used are laboratory *Pa* strain PAO1Δ*pel*Δ*psl* and PAO1Δ*mucA*[61,62], with constitutively expressed green fluorescent protein (GFP). In general, strains were grown overnight in lysogeny broth (LB) at 37 °C with shaking. M9 minimal media (Sigma Aldrich) was supplemented with 2 mM $MgSO_4$ (JT Baker) and 100 μM $CaCl_2$ (JT Baker) (henceforth referred to as M9 medium). Aggregation assays

with exogenously added polymers were performed in M9 medium (no carbon source) at 25 °C, to avoid confounding effects of growth during the assay. All biofilm growth experiments were performed in LB at 25 °C. Dextrans with molecular weights (MWs) of 6, 70, 560 and 1800–2500 kDa were purchased from Sigma-Aldrich. PSS (MW 70-1000 kDa) and PLL (MW 70 kDa) were purchased from Sigma Aldrich.

## VPS purification

VPS purification was performed according to a published protocol with several modifications[19,30]. First, a $\Delta rbmA\Delta bap1\Delta rbmC\Delta pomA$ strain was grown in LB at 30 °C overnight. 50 μL of this inoculum was added into 3 mL of LB liquid medium containing glass beads (4 mm, MP Biomedical), and the cultures were grown with shaking at 30 °C for 3–3.5 h. 50 μL of this inoculum was applied to an agar plate containing M9 medium supplemented with 0.5% glucose and 0.5% casamino acids. The plates were incubated at 30 °C for 2 days to form a continuous bacterial lawn. For each batch, 20 plates were used. The biofilms were scraped off the agar plates carefully and resuspended in 1× phosphate-buffered saline (PBS) buffer. Biofilm cells were removed by centrifugation (8000 × g, 4 °C, 45 min) and the supernatant was dialyzed for 3 days against distilled water using a dialysis cassette (10 kDa MWCO) with repeated water changes. The dialyzed sample was lyophilized to prepare crude VPS extract. The crude extract was dissolved in 10 mM Tris buffer at 1.5 mg/mL, treated with DNAse and RNAse (37 °C, 24 h), and then Proteinase K (37 °C, 48 h), followed by ultracentrifugation at 100,000 × g for 1 h to remove lipopolysaccharide. This solution was dialyzed against water for 3 days and lyophilized to provide VPS. For each purification batch, typically 10 to 15 mg of VPS was obtained as a white powder after the final lyophilization step. The VPS solutions were heated at 95 °C for 10 min to denature Proteinase K before use.

## Aggregation assay and phase diagram mapping

The aggregation assay follows a modified protocol from Secor et al.[33]. The 5Δ strain constitutively expressing mRuby3 was first cultured overnight at 37 °C with shaking in LB. After measuring the $OD_{600}$ of the culture, bacterial cells were collected and washed 3 times in M9 medium by centrifugation at 18,500 × g. Next, cells were resuspended and diluted in M9 medium (no carbon source) to an $OD_{600}$ ranging from 0.01 ($1.75 \times 10^7$ CFU/mL) to 10 ($1.75 \times 10^{10}$ CFU/mL). Concerning phase diagrams performed with chemically fixed $\Delta ABC$ cells constitutively expressing mRuby3, $OD_{600}$ of overnight bacterial cultures was measured, and resuspended to an OD = 0.01 in LB to grow statically for either 8 or 20 hours. Cells were subsequently collected and separated from VPS via centrifugation, and fixed by using a 4% paraformaldehyde (PFA) solution for 30 minutes, followed by three successive washes in M9 medium. Polymer solutions were also prepared in M9 medium. The polymer and bacterial solutions were mixed at different concentrations and transferred to individual wells of a glass-bottom 96-well plate. After sitting at room temperature (25 °C) for about 6 hours, the mixtures in the 96-well plate were imaged with a Nikon W1 spinning disk confocal microscope with a water 60× objective, using excitation at 561 nm. Note that the M9 medium contains no nutrients so no growth takes place during the aggregation assay. In mapping the phase diagram, we define an aggregate as a cluster of at least three cells close to one another. Next, we define an aggregated phase as having aggregates throughout the image, and dispersed phase as having individual cells throughout the image.

For *P. aeruginosa*, a slightly modified procedure was used. The PAO1Δ*pel*Δ*psl* strain was used because it does not produce any exopolysaccharide involved in biofilm formation[61]. The strain was first cultured overnight at 37 °C with shaking in LB. After measuring $OD_{600}$ of the culture, bacterial cells were gathered and washed 3 times in M9 (Bioworld) by centrifugation at 18 000 × g. Then, bacterial cells were resuspended and diluted in M9 to an $OD_{600}$ of 0.01, 0.04, 0.08, 0.1, 0.5,

1, 5. Alginate solutions (0, 0.05, 0.1, 0.15, 0.2, 0.5, 1, 1.5, 2, 2.5, 3, 3.5, 4, 4.5, 5, 6% wt) were prepared by serial dilution in M9 media. These polymers and bacterial solutions with different concentrations were mixed and transferred to wells of a glass-bottomed 96-well plate. After sitting at room temperature for about 3 hours, the mixtures in the 96-well plate were observed using a Nikon W1 spinning disk confocal microscope with a 60× water objective.

## Biofilm growth

All bacterial strains used were first cultured overnight at 37 °C with shaking in LB, after which the $OD_{600}$ was measured. Next, the bacterial cultures were diluted to $OD_{600} = 0.01$. 100 μL of this mixture was transferred into a well of a glass-bottomed 96-well plate and kept at 25 °C using on-stage heater. Though we used cells constitutively expressing fluorescent proteins, we found that the intensity of the fluorescent proteins under this particular growth condition was not suitable for imaging. Therefore, we generally used FM 4-64 (Sigma-Aldrich) at 2 μg/mL to stain cell membranes. When appropriate, WGA (Sigma-Aldrich) conjugated to different chromophores was used at 4 μg/mL to stain VPS. The culture was imaged using a Nikon W1 spinning disk confocal microscope.

In a subset of experiment, to demonstrate that the increase in cell aggregation during the growth of the $\Delta ABC$ strain is not due to gravity, we added Optiprep (Sigma-Aldrich) at 20, 25, 30, 35, and 45% (v/v%) to gradually increase the density of the growth medium. Optiprep is known to have minimal effect on bacterial growth. The imaging plane in this set of experiment depends on the location of the cells (Fig. S3).

For the experiments in which we manipulated intracellular c-di-GMP levels, 100 μg/mL of Kanamycin was added to LB throughout the experiment. Additionally, 1 mM of IPTG was added to LB to induce the expression of *vc1086*[44]. In a subset of experiment, IPTG was introduced during biofilm growth by adding 2 μL of IPTG (50 mM) carefully into the 100 μL culture (resulting in a final concentration of 1 mM) in the 96-well plate without disturbing the culture.

For *P. aeruginosa* biofilm, a modified procedure was used. The *P. aeruginosa* strain PAO1Δ*mucA* (a mucoid strain)[62] were cultured overnight at 37 °C with shaking in lysogeny broth (LB). After measuring $OD_{600}$ of the culture, the bacterial cultures were diluted to OD = 0.01 with LB and stained with FM 4-64 dye at 2 μg/mL. 100 μL of this mixture solution was added to a well of a glass-bottomed 96-well plate and kept at 25 °C. The 96-well plate was imaged using a Nikon W1 spinning disk confocal microscope with a 60× water objective for 24 h or every day for 5 days. To confirm the nature of the depletion aggregation, the bacterial aggregates were resuspended into solution by pipetting after phase separation was observed, and the reformation of the aggregates was recorded by acquiring time-course z-stack images every 30 s for 90 min in bright field.

## Ecological/washing experiment

After growing a bacterial culture for 12 or 17 hours, biomass quantification of the cells encased or not by VPS was performed using fluorescence confocal microscopy. Following this, a flow disturbance was introduced by gently washing the well with fresh LB medium, after which another biomass quantification was performed at the exact same *x*,*y* and *z* location. During this experiment, VPS staining was conducted using WGA-Oregon Green (4 μg/mL), and cell membrane staining was performed using FM 4-64 (2 μg/mL).

## Microscopy

Fluorescence microscopy was performed using a Yokogawa W1 confocal scanner unit connected to a Nikon Ti2-E inverted microscope with a Perfect Focus System. Cells constitutively expressing mSCFP3A, mNeonGreen, and mRuby3 were excited by lasers at 445, 488, and 561 nm, respectively, with the corresponding filter. The WGA-Oregon Green and WGA-AlexaFluor647 stains were respectively excited at 488

and 640 nm. All fluorescent signals were recorded by an sCMOS camera (Photometrics Prime BSI). Confocal images were taken using a 60× water immersion objective (CFI Plan Apo 60XC, numerical aperture = 1.20) for aggregation assays. A 100× silicone oil objective (Lambda S 100XC Sil numerical aperture = 1.35) was used for time-course, matrix staining, and ecological experiments. For mapping phase diagram and for imaging spontaneous aggregation of $\Delta ABC$ culture, a $z$-stack of 22 μm was captured with a $z$-step size of 2 μm, starting from a $z$-position of −2 to 20 μm above the glass surface. Each field of view was 220 μm × 220 μm; 3 to 4 locations were captured for each experiment as technical replicates. For staining and washing experiments, a $z$-range of 28 to 42 μm was captured with a $z$-step size ranging from 0.13 to 2 μm, starting from a z-position range of −2 to 40 μm above the glass surface. Each field of view was 130 μm × 130 μm; 3 to 4 locations were imaged for each experiment as technical replicates. During time course, imaging was performed every 30 min. For *P. aeruginosa* biofilm, the imaging volume is 135 × 135 × 32 μm with a $z$-step size of 1 μm, starting slightly below the glass. Minimal photobleaching was observed under the particular imaging condition. All images presented are raw images rendered with Nikon NIS-Elements.

## Image analysis

Characteristic length analysis: Confocal fluorescence images from a VPS-producing strain ($\Delta ABC$) at $z = 4$ μm above the glass surface obtained during time-course imaging were used for Fourier analysis to determine the characteristic size of cell aggregates ($\xi$). The raw images were first binarized before performing a 2D fast Fourier transform (FFT) using MATLAB's built-in fft2 function to obtain images in Fourier space. Spatial amplitude spectra were calculated as the radial average in the Fourier space of the absolute value of the Fourier-transformed image (Fig. S4b). The peak positions in the amplitude spectra were identified and the characteristic length of the aggregates at each time point corresponds to the inverse of the spatial frequency at the peak (Fig. S4c). The dashed line at the end of the graph (Fig. 1c) represents estimated mean data due to difficulty in obtaining precise measurements at late time.

Fraction of cells encased by VPS: Cross-sectional images obtained during time-course imaging of $\Delta rbmA$ biofilms at $z = 4$ μm above the glass surface were used to determine the fraction of cells encased by VPS. This $z$-height was chosen to minimize effect of the glass substrate while maximizing the number of quantifiable cells. In these images, the red channel representing cell areas was binarized with a fixed threshold, morphologically closed and then dilated with a circular kernel of 0.65 μm in diameter. The dilation size was chosen to match the thickness of the VPS envelope around each cell. The binary image was then used as a mask on the green channel corresponding to VPS staining. The channel that records VPS staining was further binarized to distinguish VPS straining from background fluorescence. Pixels with high intensity due to binding between WGA and cell wall in dead cells were subsequently removed. The fraction of cells encased by VPS is quantified as the ratio between the number of foreground pixels in the binarized VPS image in the mask and the total number of pixels in the mask. This quantification measures the fraction of cell area that is co-localized with VPS. The same quantification was also applied to the ecological/washing assay.

Areal fraction $\rho$: Quantification of the areal fraction $\rho$ was performed by thresholding the cross-sectional images obtained during time-course imaging of $\Delta rbmA$ biofilms at $z = 4$ μm above the glass surface. The binary area for each sample was then divided by the total area of the entire image.

Biomass quantification: The biomass quantification was performed by thresholding each 3D $z$-stack image layer-by-layer and measuring the total binarized area above the threshold in each layer.

The binary area for each sample $z$-slice was then summed to give the total biovolume, and the ratio between the total biovolume before and after the washing step was calculated.

Radial distribution function and orientational correlation analysis: 3D $z$-stacks of confocal fluorescence images were first deconvolved using Huygens 20.04 (SVI) and segmented into individual cells using methods previously described[63,64]. The radial distribution function $g(r)$ measures the probability of finding a cell at a distance $r$ from a reference cell compared to a uniform distribution and was calculated as $g(r) = \frac{n_{\text{pair}}(r)}{4\pi r^2 \Delta r} / \frac{N_{\text{pair}}}{V}$. Here $n_{\text{pair}}(r)$ is the number of cell pairs at distances between $r$ and $r + \Delta r$, and $N_{\text{pair}}$ is the total number of cell pairs in image volume $V$. $g(r)$ was further normalized by the baseline radial distribution of randomly generated positions in the same image volume to account for the finite image size. Orientational correlation was calculated as $\langle \frac{1}{2}(3\cos^2\theta_{ij} - 1)\rangle_r$, where $\theta_{ij}$ is the angle between the orientations of cells $i$ and $j$, and the angle brackets denote the average of all cell pairs at $r$. This definition yields 1 and −1/2 for aligned and orthogonal orientations, respectively, and 0 for uniformly distributed orientations.

## Growth curve

The growth curves were measured manually under conditions mimicking those during the time-course imaging. An overnight culture in LB was back-diluted in LB to yield an initial $OD_{600}$ of 0.01, and inoculated in a 96-well plate at 25 °C. For each biological replicate, 2 wells were prepared as technical replicates. Every hour, the entire culture from one of the wells was retrieved, mixed via pipetting, and $OD_{600}$ was measured. A pseudo-time course was constructed by plotting $OD_{600}$ versus the time the culture was processed. In general, two 96-well plates were used for each biological replicate.

## Protein expression and purification from *E. coli*

Bap1-GFP was purified following the procedure we published recently[30]. RbmB expression was done by cloning RbmB into a pET vector plasmid system and transformed into T7 express *E. coli* cells and cultured overnight at 37°C in LB media supplemented with 50 μg/mL Kanamycin. Expression was performed in 500 mL cultures in High-cell density growth medium. For protein purification, cells were homogenized by passing through an EmulsiFlex-C5 high pressure homogenizer and lysate cleared by centrifugation at 29,000 × *g*. RbmB was purified from the cleared lysate by loading the supernatant onto a 5 mL His-trap HF (GE Healthcare) equilibrated with high-salt TBS buffer (20 mM Tris pH 7.6, 500 mM NaCl). Column was washed with 10 column volumes of buffer containing 40 mM imidazole and protein eluted in high-salt TBS buffer containing 250 mM imidazole. Protein was further purified using a Superose S6 100/300 (Cytivia) size exclusion chromatography column in high-salt TBS buffer (20 mM Tris pH 7.6, 500 mM NaCl). Fractions containing protein was determined by SDS-PAGE and concentrated using a 50 kDa Amicon centrifugal filter.

## Electron microscopy

Digested VPS was produced by incubating 5 μg of VPS with 4 μg of RbmB at 30 °C, overnight. Bap1 (0.2 mg/mL), VPS (0.33 mg/mL), and digested VPS (0.33 mg/mL) were mixed at different ratios, and incubated for 5 mins. Prior to grid preparation, the solutions were diluted appropriately (˜100 to 1000 times) in 1× TBS. Copper grids with formvar support film (Ted Pella, Inc.) were glow-discharged for 5 mins and 4 μL of the diluted sample was spotted onto grids and incubated for 5 mins. Grids were stained 3 times with 2% uranyl acetate for 5-10 seconds per drop, in series. Excess stain was removed by filter paper and grids incubated at room temperature for 10 mins. Grids were screened using a Thermo Scientific Talos L120C transmission electron microscope.

## Bioluminescence assay for *vpsL* expression quantification

Gene expression of the *vps*-II operon was measured using a plasmid-borne luciferase reporter (pBBRlux-*vpsL*) as previously described[65]. The reporter plasmid was transferred into *Vc* strains through conjugation. The resulting reporter strains were grown in Luria Bertani broth (1% tryptone, 0.5% yeast extract, 1% NaCl at pH 7.5) with chloramphenicol (2.5 μg/mL) at 30 °C overnight, and the cultures were then diluted to $OD_{600}$ ~ 0.01. 150 μL of the two diluted cultures were added to a 96-well plate (Greiner Bio-One μClear Bottom, PS, F-Bottom, Black). Cells were incubated without shaking for 20 h at 25°C and luminescence and $OD_{600}$ were measured every 30 minutes. The results were plotted in relative luminescent units (RLU, counts $min^{-1}$ $mL^{-1}$/ $OD_{600}$). Assays were performed with at least four biological replicates. Note that it is known that measurement with plate reader tends to underestimate the OD of a biofilm-forming culture; therefore, the OD value obtained with this method is systematically lower than the manual measurement in Fig. 1d. However, the shape of the growth curves remains the same.

## Molecular dynamics simulations

We utilize coarse-grained molecular dynamics (CGMD) simulations to investigate the interaction between polymers and bacterial cells, focusing on its impact on biofilm structure. We employ the OpenFSI package by Ye and colleagues[66], designed for efficient fluid-structure interaction (FSI) simulations and implemented by Large-scale Atomic/ Molecular Massively Parallel Simulator (LAMMPS)[67] version 30 Jul 2016. The bacterial cell is simulated with a lattice membrane model with a dimension of 3 by 1 μm (length by diameter). We discretize the membrane (the bacterial surface) into a triangular network of 1179 vertices and 2354 elements, maintaining an approximately uniform Lagrangian mesh with an average size of 90 nanometers. To represent the polymers in the system, coarse-grained (CG) beads with a diameter of 100 nm are introduced to the simulation box. The mechanical characteristics of the membrane are defined by applying potential functions to the membrane's triangular networks. These functions' parameters are based on our recent work[45]. Specifics regarding these potential functions are described in Ye et al.[66]. For interparticle interactions, we use Lennard-Jones (LJ) potential with various interaction modes among the beads and nodes on the bacterial surface. The interaction details are summarized in the Supplementary Table S2.

The radial distribution function (RDF) quantifies the number density in the radial direction (*r*) between nodes from different cells. This calculation involves dividing the radial channel into uniform bins of size d*r*. Due to the anisotropic nature of the cell geometry, we calculate the node–node distance between different cells to determine the cell–cell distance instead of using the center-of-mass distance. We count the number of nodes located inside these bins and denote it as $n_{NP}$. Finally, $g(r) = \frac{n_{NP}}{VN_{nodes}}$ is computed, where *V* represents the bin volume and $N_{nodes}$ represents the total number of nodes in the channel. The appearance of the peak at 1.14 μm in the RDF profile indicates a strong parallel alignment of the rod-shaped bacterial cells. This feature served as a criterion to distinguish between a depletion-aggregated and a well-dispersed system. When the cells are bridged by polymers, the sharp peak at 1.14 μm in the RDF profile is replaced by a broader peak with the peak position shifted to a higher value. This occurs because bridging increases the distance between cells, resulting in a higher cell-to-cell separation.

The bacterial cells are randomly positioned within a simulation box of 6.2 × 6.2 × 6.2 $μm^3$, in which periodic boundary conditions are applied in the *x*, *y*, and *z* directions.

To systematically study the depletion behavior of bacterial cells (Fig. 2g and S7a), we vary three parameters: the number of cells, the number of polymer beads, and VPS surface coverage. VPS surface coverage is adjusted by randomly replacing native cell surface beads with VPS-coated surface beads. The interaction between VPS beads and cellular surface nodes depends on the type of node: native cell surface nodes exhibit repulsive interaction with the polymer beads (1 $k_BT$), while VPS-coated surface nodes have a much weaker repulsion (0.1 $k_BT$). The number of cells ranges from 6 to 45, corresponding to volume fractions from 5.4% to 39.4%. The number of VPS polymer beads varies between 30,000 and 100,000, representing volume fractions of 6.6% to 21.9%. The percentage of VPS-coated beads ranges from 0% (no VPS coating) to 100% (full VPS coating). Input files to reproduce our 3D phase diagram can be found in the corresponding folder in the public Github https://github.com/DanhNguyen-UWMad/ Bacteria-Depletion/tree/main/Fig2g.

To investigate the impact of VPS bead properties on the depletion process (Fig. S7b–d), we use a model of 12 bacterial cells interacting with 80,000 VPS beads. We switch the interaction between the cell surface beads and the VPS beads from repulsion to strong attraction with an attraction strength of 50 $k_BT$. Additionally, the interaction between the VPS beads is also changed from pure repulsion to strong attraction, allowing us to observe the effect of VPS bead properties (attractive or repulsive) on cell aggregation. To incorporate the attractive interactions, we extend the cut-off distance for the LJ potentials to 2.5 in the simulations. We also vary different attraction strengths (18 $k_BT$, 30 $k_BT$, and 50 $k_BT$) to study the effect of attraction strengths on cell bridging (Fig. S7e). To examine the effect of VPS polymer bead concentrations on bridging aggregation, we reduce the number of attractive VPS beads from 80,000 to 25,000 (Fig. S7f). Finally, to achieve a transition between depletion and bridging, we randomly select a portion of the VPS beads (up to 20%, or 16,000 beads) in the system containing 12 cells and 80,000 VPS beads and convert them to sticky beads. These sticky beads can attract both the cell surface beads and each other while remaining repulsive to the native VPS beads. The disappearance of the sharp peak at 1.14 μm in the RDF profiles, which signifies the depletion of bacterial cells, is used to track the transition between depletion and bridging (Fig. 4d).

In all simulations, we employ the Langevin thermostat to model the interaction of bacterial cells and polymers with a background implicit solvent. This thermostat is combined with the NVE (conserving number of atoms – volume – total energy) ensemble to perform Brownian dynamics with the damping factor of 100. The temperature of the system is maintained at 300 K in all models. For each simulation, the cells and polymer beads are first randomly distributed in the simulation box, followed by the relaxation process lasting 10,000,000 steps prior to a long production run. To ensure the phase separation is completed, the long production run for each model is performed with a maximum of 100,000,000 running steps. The timestep used in all the models is set at 0.01 (11.4 ns). As a result, the maximum runtime for a depletion model is calculated at 100,000,000 × 11.4 ns = 1.14 s. However, in most cases, cells rapidly aggregate within 20,000,000 steps if the depletion condition (i.e., the number of polymer beads is sufficient) is satisfied. Therefore, this coarse-grained simulation setup is computationally feasible without the need for enhanced sampling methods. The last configuration of the system at the timestep of 100,000,000 will be used to analyze the RDF. We perform three independent runs for each simulation condition to ensure consistent observations across multiple runs. The outputs of each run are available on our GitHub repository. To assess the influence of initial configurations on the depletion process, we perform three independent runs with varying initial configurations, specifically for the case of 12 cells interacting with 80,000 repulsive polymer beads (https://github.com/DanhNguyen-UWMad/Bacteria-Depletion/ tree/main/Fig4d/0Per). These runs demonstrate similar depletion outcomes. All the general parameters of CG modeling are summarized in Table S2. All snapshots of CGMD simulations are rendered using VMD[68] version 1.9.3. Input files to reproduce our results can be found in the corresponding folder in our GitHub repository: https://github. com/DanhNguyen-UWMad/Bacteria-Depletion/tree/main.

## Statistical analysis

Errors correspond to standard deviations (SDs) from measurements taken from distinct samples. Standard $t$-tests were used; the specific comparisons and results are indicated in each figure caption. Tests were always two-tailed and paired or unpaired depending on the details of the experimental design. All statistical analyses were performed using GraphPad Prism software.

## Reporting summary

Further information on research design is available in the Nature Portfolio Reporting Summary linked to this article.

## Data availability

Source data are provided as a Source Data file with this paper. Raw microscopy data generated in this study have been deposited in Dryad under accession code https://doi.org/10.5061/dryad.zcrjdfnph. Source data are provided with this paper.

## Code availability

MD simulation codes used in the manuscript have been deposited in Github under accession code https://github.com/DanhNguyen-UWMad/Bacteria-Depletion/tree/main.

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

## Acknowledgements

The research reported in this publication was supported by the National Science Foundation awarded to J.Y. and Y.L. (DMR/MPS 2205006 and 2313746) and to V.G. (CMMI 2150878). This work is also supported by the National Institute of General Medical Sciences awarded to R.O. (R15GM152959) and to S.S. (K12GM133314), and by the National Institute of Allergy and Infectious Diseases awarded to W.N. (R01 AI121337). C.D.N. received support from the Simons Foundation (award number 826672), NSF IOS grant 2017879, and NIGMS grant 1R35GM151158-01. We acknowledge Dr. Christopher Waters for providing the plasmid encoding VC1086. We thank Dr. Jun Liu, Dr. Shuaiqi Guo, and Mr. Botting for the experimental help. We thank Dr. Fitnat Yildiz for providing insightful suggestions. We acknowledge help from Dr. Kaifeng Zhou from Yale Science Hill Cryo-EM core for assistance in electron microscopy. Imaging of *P. aeruginosa* was done at the Center for Biomedical Research Support Microscopy and Imaging Facility at UT Austin (RRID:SCR_021756). N.K. thanks the support of NSF grant DMS 2435484 and is gratefully acknowledged.

## Author contributions

A.M. and J.Y. conceptualized the project. A.M. and J.Y. performed the experiments and data analysis in *Vc* biofilms. K.M. and J.Y. constructed the *Vc* strains. J.S.T. performed single-cell analysis. A.H., R.W., and R.O. provided the purified proteins. D.T.N. and Y.L. performed MD simulations and associated analysis. X.Z. and V.G. performed the experiments in *P. aeruginosa*. S.S. and W.N. performed *vpsL* expression quantification. A.Z., I.R.B., and N. K. performed the stochastic modeling. C.D.N provided valuable feedback on the ecological significance of the experiments. A.M. and J.Y. drafted the manuscript and all authors contributed to the final manuscript.

## Competing interests

The authors declare no competing interests.
