## [Peer review file · Nature Communications]

Surface remodeling and inversion of cell-matrix interactions underlie community recognition and dispersal in *Vibrio cholerae* biofilms

Corresponding Author: Professor Jing Yan

Version 0:

Reviewer comments:

Reviewer #1

(Remarks to the Author)

This manuscript reports measurements addressing the hypothesis that the morphology of *Vibrio Cholera* biofilms is dictated by the physical mechanisms of depletion-induced and bridging-induced aggregation. *Vibrio* biofilms adopt states dominated by one or the other of these aggregation mechanisms, depending on whether the *vibrio* biofilm polysaccharide VPS is locally available on the cell surface. Cell surface availability of VPS is in turn dictated by whether the cell is synthesizing VPS or not, and this is growth phase dependent.

To support this hypothesis, the paper presents evidence that: (1) *vibrio* cell devoid of VPS aggregate through depletion (Figure 1); (2) *vibrio* shows two growth phases, the first is characterized by VPS production while the second is not. At the same time, the first growth phases yields bridging aggregated biofilms, while the second phase yields depletion aggregated biofilms (Figure 2); (3) proteins like BAP1 crosslink both surface and matrix VPS, thus promoting bridging, but hindering depletion (Figure 3); (4) enzymes such RbmB cleave VPS from the cell surface, thus inhibiting the bridging mechanism (Figure 5); (5) washing eliminates VPS-free cells from *Vibrio* biofilms (Figure 6).

Although there have been papers that have addressed the physics of planktonic to bridging biofilms (e.g. ref 8) and planktonic to depletion biofilms (e.g. ref 21), I am not aware of a paper addressing a possible transition between these two mechanisms of biofilm formation. This is therefore a strong claim of significance, and I also believe that those claims are largely validated by the paper, although with some questions described below. I am less convinced of the significance and validity of the claims about dispersal. I also think the paper fails in its claim that the mechanism can be generalized to other species, because the *pseudomonas* section of the paper addresses neither the transition between the two types of aggregation mechanism nor dispersal.

The authors should address the following specific comments during revision if this paper for the paper to be considered further by this journal.

1) The central claim of the biofilm formation part of the paper is that bare cells and VPS do not form a biofilm by bridging aggregation; the cells behave like systems in which polymer and cell are of like charge, which does not support bridging. Bridging aggregation biofilms instead form when cells are covered with VPS. Here I think the paper underestimates the extent to which proteins like Bap1 must play a role and ignores the role of entanglement and association in VPS. Physically, I see no reason that a VPS surface coated cell/VPS matrix system should aggregate without these interactions: a VPS coated cell is cell negatively charged, and therefore no more available to aggregate with the VPS matrix than a bare cell. This would leave only physical entanglements or transient associations to support the interface between the surface bound VPS and the matrix VPS, or some link of protein induced crosslinking or association. How the VPS matrix generates its physical integrity, and how the interface between cell and matrix is supported in the bridging case is insufficiently addressed in the paper. That is, entanglement, associative, and protein mediated interactions of the polysaccharide are not implicated in the hypothesis, but would seem relevant, at least in the bridging case.

2) I do not see evidence that the dispersal mechanism can occur in the absence of interactions generated by proteins such as RbmB; consequently, I do not see new physics beyond the ideas advanced in earlier publications, especially refs 13 and 14. That is, what is described here appears essentially as in these papers. That is, an enzyme like RbmB cleaves VPS from

the surface, thereby dispersing cells that are bridged. Although depletion aggregated cells could be released when VPS concentration drops below the concentration needed to sustain that phase, the key evidence of Figure 6c doesn't seem to distinguish if these cells are freed of VPS through an enzymatic mechanism or not. The introduction claims that the mechanisms of ref 13 and 14 are incomplete because dispersal can happen rapidly (less than 30 minutes), and therefore non-enzymatically, but that kind of distinction is not well supported by Figure 6. More care to discuss how these results represent a new picture of dispersal relative to these prior papers is needed.

3) The section on pseudomonas does not support generalizability of this paper's findings beyond vibrio. It, in my view, does no more than establish depletion aggregation in a second species, a claim already available by comparison with ref 21. There is no evidence to support the key new claims of this paper, namely the transition from one aggregation type to the other based on growth phase, and dispersal. I suggest this section be removed and the claims of the paper revised to be limited to vibrio.

4) Likewise, how the simulation and modeling of the paper supports the claims of the paper is unclear. The molecular dynamics section is disconnected from the experiments because the VPS/surface binding and cell VPS/matrix VPS interaction isn't modeled. Polysaccharide associations typically require much more sophisticated modeling than applied here, because of the associative and entanglement interactions.

5) In the section of lines 282-296, it is not very clear how to link the RbmB role with the findings of this section. Could the authors please address this?

6) Please clarify the strains used in the section beginning at line 309.

Reviewer #2

(Remarks to the Author)

The manuscript from Dr. Yan and co-workers describes, in detail, how extracellular polysaccharides and associated proteins in *Vibrio cholerae* strains determine colony structure. Specifically, the authors ask the question whether the main polysaccharide VPS is by itself an adhesin that drives aggregation of individual cells. Using microscopy to analyze cell aggregation for an extensive set of mutant strains, they are able to show that VPS drives formation of compact biofilm structures via depletion forces. Inducible expression of a phosphodiesterase that degrades c-di-GMP allows the authors to demonstrate directly that abolishing VPS production suppresses the depletion-aggregation process. Different types of stochastic simulations are used to model this transition in silico. In addition, the role of further core matrix proteins is carefully analyzed. Trimming of cell surfaces by RbmB leads to remodeling of the biofilm and shifts the interaction from an attractive one to repulsion.

The beauty of this manuscript lies in its combination of a materials-engineering perspective with in-depth understanding of the microbiology at hand. A wealth of novel data is presented. The interpretations are solid and the article is well-written. The discovery of a generic role of depletion forces for bacterial colony formation, along with biological mechanisms for its regulation, is clearly of broad interest for researchers from different fields, including biology, biophysics and engineering. Publication is recommended.

Comments:

-The authors repeatedly state that it is "commonly assumed that biofilm-dwelling cells are glued together by the matrix". While this formulation may have indeed been used by a number of authors, most researchers in the field are aware of the complex chemical properties of biofilm matrix molecules. Therefore, I think, the manuscript does not correct a commonly held misperception. The work goes far beyond that and provides a wealth of new data and fundamental new insights.

-The slope of the phase diagrams (concentration of bacteria vs polymer concentration) is used to distinguish between two assumed types of interactions, namely depletion forces and crossbridge formation. While this interpretation is appealing, the manuscript would benefit from an extended discussion of analogous colloidal systems.

-Depletion interactions and direct coupling of polymers and cells (either as cross bridges or to individual cells) may occur both simultaneously. Can one disentangle the two effects in simulations?

-The impressive MD simulations produce beautiful data. However, the connection between experimental data and the simulation results could be strengthened. Perhaps, one could show experimental RDF data, which the authors likely have already collected...

-Possibly, extracellular addition of Bap1 has been studied in earlier experiments since this molecule is a well-studied factor.

-Figure 6 shows a very neat combination of simulations and experimental data. It is demonstrated that, under flow, cells encased by VPS largely remain in place, whereas cells in the depletion aggregates and not encased by VPS are removed by shear forces. It is suggested that in this process excludes non-VPS producing cells. Here, it appears unclear from the data if these cells really do not produce VPS. Is it possible that VPS is just naturally distributed inhomogeneously, with some cells being covered less than others although producing as much VPS? A direct measurement of VPS production in cells could also improve the simulations since the parametrization could be improved.

-A second comment on Fig. 6 regards speculation about the ecological function of such a process: sometimes, dispersal can be advantageous. Expelling cells that do not produce VPS may only be advantageous from the perspective of the existing colony. If daughter colonies produce less VPS, since they are offspring of “cheaters”, the process could reduce in the overall fitness of the species.

-The measurements on the role of alginate for depletion interaction of *P. aeruginosa* are very interesting and indeed generalize the finding for *V. cholerae*. However, it should be explained also in the main text that a PAO1 Δ pel Δ psl is used for the experiments. This is a very particular mutant.

Minor:

- A couple of typos: Course-grained simulation -> Coarse ...

Reviewer #3

(Remarks to the Author)

This is an interesting study that focusing on analysing the cell-matrix interactions during biofilm-aggregate formation, using the model organism *Vc*. Authors present some interesting results; I particularly like the phase diagram analyses. However, my main concern is that in the current form, a number of conclusions are not supported by the data, as presented. My specific comments are below.

Major comments

1. As authors are most likely aware, the biofilm field is shifting to analyse biofilms in models that more closely represent the natural environments that these organisms encounter. This study is highly artificial, both in the media used to grow the biofilm-aggregates (LB or M9; this was unclear), and that only mutants were analysed. It is unclear if these observations can be replicated with WT cells, or in nutritional conditions that mimic in vivo environments. It is therefore unclear how relevant these observations are to the ecology of the organism, outside of this artificial lab conditions.
2. Authors make a lot of claims regarding VPS configurations in the matrix, i.e VPS coating *Vc* cells, being crosslinked by Bap1. However, currently these conclusions are not supported by the data, either due to a lack of quantification or microscopy images that are unclear. Imaging VPS in the matrix (with cells) by super resolution microscopy would support these conclusions.
3. Due to the mutants used across the different assays it is difficult to compare results. *5* doesn't produce matrix components, *ABC* doesn't produce matrix proteins, but overproduces VPS, while *rbmA* *rbmB* produces WT amounts of VPS. This complicates interpretation.
4. Authors may want to consider determining if the *5* mutant has the same surface change, LPS structure etc compared to WT. This would be an important control as a number of surface structures have been deleted which may introduce artifacts into the results.
5. Figure 1a. Are the top two fluorescent images zoomed insets of the lower image? Is the top image of *5* incubated with chitosan and middle image of *5* incubated with pVPS? What is the lower image? Clarifying this in the legend would be helpful, as this was not clear.
6. Figure 1a and S2. Assuming that the CFU value on the x-axis is the amount of bacteria added to the polymer? Or was this the amount after the 6h incubation? This was not clear.
7. Figure 1b. Do authors see the same phenotype in *ABC*, without the DGC mutation? It would be interesting to determine if overproduction of VPS is required for this phenotype.
8. Figure 1b. Authors state that depletion aggregation is occurring. However, from the representative images it is unclear what the cellular arrangements are. I would suggest providing clearer images to support this conclusion.
9. Figure 1d. How did authors control for aggregation reducing the OD values, by the aggregates settling out of solution? Would CFU/mL be a more accurate measurement.
10. Figure 2a-c. I am confused by this data, and the conclusions drawn from it. Couldn't the differences be attributed to comparing VPS overproducing cells (*ABC*) to non-VPS producing cells (*5*)?
11. L158-159. What is the evidence for this?
12. L163. Authors state that *ABC+RbmB* have a phase diagram similar to *5*. However in Figure 2c the solid (*ABC+RbmB*) and dashed (*5*) lines are very different. Please clarify this statement.
13. Quantifying VPS levels, or staining VPS and visualizing alone with the cells by microscopy throughout the experiments, especially for the *ABC* and *ABC+RbmB* experiments, would help support the conclusions.
14. L173. Authors states that induction of VC1086 abolishes VPS production. However, there is no quantification of VPS levels. This is an overinterpretation of the data. Furthermore, the images presented in Figure 2d as evidence of loss of aggregation are difficult to interpret. Both appear to be a lawn of cells.
15. L177 – 183. These conclusions are not supported by the data. Authors have not quantified either c-di-GMP or VPS levels in these experiments.
16. L207 – 208. This experimental set up was confusing. Assuming purified VPS and Bap1 were mixed together? VPS and Bap1 alone controls need to be included. The electron microscopy images in Figure S7 are unclear.
17. Figure 3b. For the phase diagrams in this panel and throughout the other figures, how did authors define an aggregate? Was there a minimum number of cells incorporated? Authors state that in this figure, no aggregates were observed. However, in the image provided there appears to be two aggregates. Also Bap1 should also be labelled to confirm the reported conclusions, that Bap1 crosslinks with VPS, preventing aggregation.
18. L218 – 228. Authors make claims regarding Bap1 localization, however Bap1 is not labelled in any of the microscopy images. Therefore, authors have over interpreted their results.

19. L243. From the images provided in Figure 4 it is difficult to observe the cellular arrangement that supports the claim that aggregates are formed by depletion.
20. Figure S8. Authors state that RbmC abolished bridging aggregation. However in the provided image a large aggregate is present, that appears to have a clumped cellular arrangement. Can authors clarify this statement.
21. Figure S9. The conclusion that 5 cells are excluded from the aggregates is unclear. This should be confirmed with imaging analysis. The fact that there appears to be few 5 cells further confound the results. Furthermore, L289 authors state that expelled cells form depletion aggregates. However the 5 cells in Figure S9 are single cells, and do not support this conclusion.
22. Figure 6. What was the rationale for using *rbmA* mutant for these experiments?
23. L403 – 404. Authors state that aggregation assays were performed in M9, and biofilm experiments were performed in LB. However, all experiments in the manuscript focused on aggregation. It is unclear what growth conditions were used.

Reviewer #4

(Remarks to the Author)

The manuscript of Moreau et al. investigated bacteria-matrix interaction in *Vibrio cholera* and suggesting a new conceptual. In general, the data and model can support the conclusion although the logic is no very easy to follow.

Main constructive criticism :

1. Introduction should include more information, such as the current understanding about the function of VPS, RbmABC and Bap1 as well as their contribution in the matrix; How C-di-GMP affect the expression of these genes, etc. These will help reader to understand the results.
2. What is the function of *pomA*? I did not find any information about this gene in entire manuscript.
3. Is VPS a secreted exopolysaccharide or bacterial surface associated? If this is the first study about the localization of VPS, the author should provide more data.
4. L118, How the *vpvC* mutant affect the C-di-GMP level, increase or decrease?
5. L238, Why use *rbmA* deletion mutant? If in a *RbmA+* background, what would be the results?
6. L259, Does RbmB release VPS or degrade it?
7. Fig.2, need the control of vector -IPTG for 13h,+IPTGfor 12h
8. Fig.4, How to determine the phase I and II? There is a strong signal of VPS at 4.5h, but little at 13h. Where those VPS has gone? Be degraded or removed by staining processing?
9. For *Pseudomonas aeruginosa*, Psl polysaccharide is more important for biofilm formation and cell-cell interaction. Mucoid strain also requires Psl to form biofilm. I am wondering whether the depletion-aggregation occurs in PAO1.

Reviewer #5

(Remarks to the Author)

Version 1 :

Reviewer comments:

Reviewer #1

(Remarks to the Author)

I have read the revised manuscript and the authors' rebuttal letter. I think the authors have appropriately responded to both my comments and those of the other reviewers. More specifically, the additional simulations and shift of the *pseudomonas* section to the discussion improve the manuscript. I also think the revision is more nuanced in its description of the claims of the paper, and provides better connection to the prior literature. I am pleased to support publication.

Reviewer #2

(Remarks to the Author)

Although the original manuscript was already of a high standard, the authors have invested significant effort into the revision process, resulting in a manuscript that is even more robust and compelling.

A few remarks are in order:

The experiments are based on a particular choice of bacterial wt strains and mutants, yet they are not excessively "artificial" in my opinion. Indeed, the controlled conditions that the authors expertly used to prepare their system allow them to answer their fundamental questions about mechanisms. This is an excellent example of basic science at its best.

Furthermore, the authors have conducted additional experiments to demonstrate that the physics of the aggregation phenomenon observed in this study remains valid for WT cells in constrained environments. Given that growth occurs in constrained environments in numerous real-world niches of the pathogen, I believe that this new supplementary information

effectively illustrates the physiological relevance of the work.

The newly added systematic simulation results regarding the behaviour of the radial distribution function with varying polymer fraction in Fig. 4 provide a valuable complement to the experimental data. Furthermore, the authors have now quantified the RDF along with the orientation correlation in their experiments. The additional data demonstrating that the peak in the RDF occurs at a significantly larger distance in the bridging case compared to the depletion case provides further indirect proof of the different physical nature of these two interactions.

A quantification of the amount of VPS would undoubtedly be beneficial in all mutants. Despite technical challenges, the authors have employed a luminescence reporter to quantify the expression of VPS as a function of growth phase, thereby providing the relevant information.

The authors have fully exploited the opportunity to enhance the clarity of the manuscript and to address all comments from the reviewers during the revision process.

Reviewer #3

(Remarks to the Author)

Authors have successfully address all of my previous concerns/ comments
Great manuscript!

Reviewer #4

(Remarks to the Author)

All my questions have been addressed.

Reviewer #5

(Remarks to the Author)

Response to Reviewers

Following the reviewers' comments, we have made significant revisions to the manuscript, including additional experimental and simulation results. We have also added 4 new panels to main figures and 7 new SI figures in response to the reviewers' requests. Point-by-point responses to the reviewers' comments are listed below, where the reviewers' comments are in black, and our responses are in blue. We have also highlighted all major changes in the text.

Reviewer #1:

This manuscript reports measurements addressing the hypothesis that the morphology of *Vibrio Cholera* biofilms is dictated by the physical mechanisms of depletion-induced and bridging-induced aggregation. *Vibrio* biofilms adopt states dominated by one or the other of these aggregation mechanisms, depending on whether the vibrio biofilm polysaccharide VPS is locally available on the cell surface. Cell surface availability of VPS is in turn dictated by whether the cell is synthesizing VPS or not, and this is growth phase dependent.

To support this hypothesis, the paper presents evidence that: (1) vibrio cell devoid of VPS aggregate through depletion (Figure 1); (2) vibrio shows two growth phases, the first is characterized by VPS production while the second is not. At the same time, the first growth phases yields bridging aggregated biofilms, while the second phase yields depletion aggregated biofilms (Figure 2); (2) proteins like BAP1 crosslink both surface and matrix VPS, thus promoting bridging, but hindering depletion (Figure 3); (3) enzymes such RbmB cleave VPS from the cell surface, thus inhibiting the bridging mechanism (Figure 5); (4) washing eliminates VPS-free cells from *Vibrio* biofilms (Figure 6).

Although there have been papers that have addressed the physics of planktonic to bridging biofilms (e.g. ref 8) and planktonic to depletion biofilms (e.g. ref 21), I am not aware of a paper addressing a possible transition between these two mechanisms of biofilm formation. This is therefore a strong claim of significance, and I also believe that those claims are largely validated by the paper, although with some questions described below. I am less convinced of the significance and validity of the claims about dispersal. I also think the paper fails in its claim that the mechanism can be generalized to other species, because the pseudomonas section of the paper addresses neither the transition between the two types of aggregation mechanism nor dispersal.

The authors should address the following specific comments during revision if this paper for the paper to be considered further by this journal.

Response: We thank the reviewer for the positive assessment of our manuscript. We particularly appreciate that the reviewer points out the novelty of our findings. We clarified in the *Pseudomonas* session (now at the end of the Discussion) that, in the case of mucoid *Pa* strains, we have only showed the possibility of spontaneous depletion aggregation; we have correspondingly tuned down the text in relevant places.

1) The central claim of the biofilm formation part of the paper is that bare cells and VPS do not form a biofilm by bridging aggregation; the cells behave like systems in which polymer and cell are of like charge, which does not support bridging. Bridging aggregation biofilms instead form when cells are covered with VPS. Here I think the paper underestimates the extent to which

proteins like Bap1 must play a role and ignores the role of entanglement and association in VPS. Physically, I see no reason that a VPS surface coated cell/VPS matrix system should aggregate without these interactions: a VPS coated cell is cell negatively charged, and therefore no more available to aggregate with the VPS matrix than a bare cell. This would leave only physical entanglements or transient associations to support the interface between the surface bound VPS and the matrix VPS, or some link of protein induced crosslinking or association. How the VPS matrix generates its physical integrity, and how the interface between cell and matrix is supported in the bridging case is insufficiently addressed in the paper. That is, entanglement, associative, and protein mediated interactions of the polysaccharide are not implicated in the hypothesis, but would seem relevant, at least in the bridging case.

Response: We thank the reviewer for pointing out the importance of the potential role of entanglement and VPS self-association in the system. Indeed, for long polysaccharides, entanglement and self-association are important potential contributors to matrix integrity and mechanics, as shown in other species such as *Staphylococcus epidermidis* (PMID: 23540609). In the case of *V. cholerae* (*Vc*), unfortunately, too little is known about VPS regarding its physiochemical properties, including molecular weight, persistent length, polymer-polymer interactions etc, to generate a quantitative assessment of the contribution of these factors. We are performing these characterizations in ongoing projects. We have added a dedicated paragraph in the Discussion session regarding this aspect.

Regarding the current manuscript, we showed that in the case when Bap1 (or more precisely the β -propeller domain) is present, it may crosslink VPS both in the secreted form and the surface-associated form, and this crosslinking is critical to bridging. When Bap1 and RbmC are absent, i.e., in the ΔABC mutant, in the first growth phase, as the reviewer suggests, the negatively charged cells are indeed just suspended in the loose network of VPS, which, likely is entangled or self-associated due to hydrogen bonding interactions. We didn't go into details of the nature of the uncrosslinked VPS network since we do not have the relevant information yet.

2) I do not see evidence that the dispersal mechanism can occur in the absence of interactions generated by proteins such as RbmB; consequently, I do not see new physics beyond the ideas advanced in earlier publications, especially refs 13 and 14. That is, what is described here appears essentially as in these papers. That is, an enzyme like RbmB cleaves VPS from the surface, thereby dispersing cells that are bridged. Although depletion aggregated cells could be released when VPS concentration drops below the concentration needed to sustain that phase, the key evidence of Figure 6c doesn't seem to distinguish if these cells are freed of VPS through an enzymatic mechanism or not. The introduction claims that the mechanisms of ref 13 and 14 are incomplete because dispersal can happen rapidly (less than 30 minutes), and therefore non-enzymatically, but that kind of distinction is not well supported by Figure 6. More care to discuss how these results represent a new picture of dispersal relative to these prior papers is needed.

Response: We thank the reviewer for pointing out the confusion in the original manuscript. Indeed, consistent with Refs. 13 and 14, our results still show that the enzymatic activity of RbmB is needed for complete dispersal. However, unlike previous reports (PMID:33288715 and bioRxiv <https://doi.org/10.1101/2024.07.15.603607>) which assume that RbmB enables cell dispersal via matrix degradation, in the current manuscript we suggest the trimming of surface-anchored VPS

is sufficient and perhaps is the dominant mechanism for its biofilm-dispersing activity. It is still unclear where RbmB is located in the biofilm (intra or extracellular), but the surface trimming mechanism we described here offers a mechanism for RbmB to enable cell dispersal regardless of whether it is secreted. We are not aware of any prior report that explicitly made this point. We also agree with the reviewer that depletion aggregation isn't a prerequisite for dispersal; surface remodeling and consequent inversion of cell-matrix interaction are. We have significantly updated the text, and we have added relevant discussions at the end of the manuscript.

3) The section on pseudomonas does not support generalizability of this paper's findings beyond vibrio. It, in my view, does no more than establish depletion aggregation in a second species, a claim already available by comparison with ref 21. There is no evidence to support the key new claims of this paper, namely the transition from one aggregation type to the other based on growth phase, and dispersal. I suggest this section be removed and the claims of the paper revised to be limited to vibrio.

Response: In response to the reviewer's concerns, we significantly tuned down the claims in the *Pseudomonas* session. Instead of a standalone result session, we have merged it with Discussion in the updated manuscript. We have also confined our claims about *Pa* biofilm to the possibility of spontaneous depletion aggregation.

4) Likewise, how the simulation and modeling of the paper supports the claims of the paper is unclear. The molecular dynamics section is disconnected from the experiments because the VPS/surface binding and cell VPS/matrix VPS interaction isn't modeled. Polysaccharide associations typically require much more sophisticated modeling than applied here, because of the associative and entanglement interactions.

Response: In response to the reviewer's suggestion, we performed additional simulations to strengthen the connection between experimental and simulation results. Specifically, we modeled the inversion of the cell-matrix interactions starting from the bridging case (100% attractive polymer beads) and then gradually increasing the fraction of polymer beads that are repulsive to each other and to the cell surface. Results are shown in the updated Fig. 4d. Briefly, with this simulation scheme, we were able to capture the transition from bridging to depletion, as we saw experimentally. We have also varied the attraction strengths in the simulation (Fig. S7e) and showed that a strong attraction ($50 k_B T$) gives cluster morphologies similar to the experimental data, in which cells are randomly oriented and also bridged by polymers. This potentially suggests that the cell-cell interaction conferred by VPS crosslinking in the experiment is also strong compared to thermal energy; indeed, we do not see much thermal fluctuation of cells in the biofilm. We have included these discussions in the updated manuscript.

We also agree with the reviewer that polysaccharide association is a complicated process, and a more sophisticated modeling approach is needed for studying the polymer systems *per se* – indeed, MD simulations treating the polymers as flexible chains crosslinked by Bap1 are underway in the lab. However, these sophisticated models are computationally expensive and therefore incompatible, at least at the current stage, with the simulation of a full biofilm with cells – this is a common constraint for multiscale systems. Therefore, as a first step to simulate this complex system, we chose to grossly simplify the polymer molecules into beads that are attractive or repulsive to each other and to cell surfaces mimicking the effect of depletion and crosslinking, in

the current manuscript.

5) In the section of lines 282-296, it is not very clear how to link the RbmB role with the findings of this section. Could the authors please address this?

Response: We have clarified in this section that the role of RbmB is primarily to trim VPS off cell surfaces and consequently, facilitate the conversion of cell-matrix interactions from attractive to repulsive during nutrient starvation and therefore, dispersal.

6) Please clarify the strains used in the section beginning at line 309.

Response: Done.

Reviewer #2:

The manuscript from Dr. Yan and co-workers describes, in detail, how extracellular polysaccharides and associated proteins in *Vibrio cholerae* strains determine colony structure. Specifically, the authors ask the question whether the main polysaccharide VPS is by itself an adhesin that drives aggregation of individual cells. Using microscopy to analyze cell aggregation for an extensive set of mutant strains, they are able to show that VPS drives formation of compact biofilm structures via depletion forces. Inducible expression of a phosphodiesterase that degrades c-di-GMP allows the authors to demonstrate directly that abolishing VPS production suppresses the depletion-aggregation process. Different types of stochastic simulations are used to model this transition *in silico*. In addition, the role of further core matrix proteins is carefully analyzed. Trimming of cell surfaces by RbmB leads to remodeling of the biofilm and shifts the interaction from an attractive one to repulsion.

The beauty of this manuscript lies in its combination of a materials-engineering perspective with in-depth understanding of the microbiology at hand. A wealth of novel data is presented. The interpretations are solid and the article is well-written. The discovery of a generic role of depletion forces for bacterial colony formation, along with biological mechanisms for its regulation, is clearly of broad interest for researchers from different fields, including biology, biophysics and engineering. Publication is recommended.

Response: We sincerely thank the reviewer for the encouragement and appreciation of our results.

-The authors repeatedly state that it is “commonly assumed that biofilm-dwelling cells are glued together by the matrix”. While this formulation may have indeed been used by a number of authors, most researchers in the field are aware of the complex chemical properties of biofilm matrix molecules. Therefore, I think, the manuscript does not correct a commonly held misperception. The work goes far beyond that and provides a wealth of new data and fundamental new insights.

Response: We thank the reviewer for clarifying this point. We have updated the text in the introduction accordingly.

-The slope of the phase diagrams (concentration of bacteria vs polymer concentration) is used to distinguish between two assumed types of interactions, namely depletion forces and crossbridge formation. While this interpretation is appealing, the manuscript would benefit from an extended discussion of analogous colloidal systems.

Response: We have added a paragraph in the Discussion session following the reviewer's suggestion.

-Depletion interactions and direct coupling of polymers and cells (either as cross bridges or to individual cells) may occur both simultaneously. Can one disentangle the two effects in simulations?

Response: In response to the reviewer's suggestion, we have performed additional simulations to disentangle these two effects. Specifically, we modeled the inversion of the cell-matrix interactions by gradually increasing the fraction of polymer beads that are repulsive to each other and to the cell surface, starting from the bridging case (100% attractive polymer beads). Results are shown in the updated Fig. 4d. Briefly, we found that the contributions from the attractive and repulsive beads are asymmetric: a small fraction (~1%) of attractive beads can induce bridging aggregation, which is indicated by the shift of the peak position away from the close contact value in simulation (1.14 μm). In the experiment, both depletion and bridging effects are present when the crosslinkers Bap1/RbmC are present, but given the asymmetry seen in the simulation, we expect the bridging mechanism to dominate unless the cell surface is devoid of VPS.

-The impressive MD simulations produce beautiful data. However, the connection between experimental data and the simulation results could be strengthened. Perhaps, one could show experimental RDF data, which the authors likely have already collected...

Response: In response to the reviewer's question, we have performed single-cell resolution imaging of the aggregates, for both depletion- and bridging-aggregates. We have also performed the corresponding RDF characterization for the cell centroids (updated Fig. 3d). We do observe significant differences in the two systems, both in terms of cell-cell distance and the coupling of cell orientation: in the case of depletion, there is a strong correlation between neighboring cells indicative of parallel alignment, and the RDF has a peak at 0.95 μm . In the case of bridging, cell orientation is uncoupled between neighboring cells. Moreover, the RDF is broader with a peak at a much larger distance (2.35 μm), due to the random cell orientation and the polymers between the neighboring cells.

-Possibly, extracellular addition of Bap1 has been studied in earlier experiments since this molecule is a well-studied factor.

Response: In fact, the biochemistry of Bap1 has only been recently revealed by our team (Kaus *et al. J. Bio. Chem.* 2019; Huang *et al. Nat. Commun.* 2023). An exogenous addition experiment of Bap1 has been performed in Absalon *et al. PLoS Pathog* 2014. However, these previous manuscripts have been focused on the adhesive function of Bap1 (and RbmC); here, we emphasize their role in VPS crosslinking and in controlling cell-matrix interaction.

-Figure 6 shows a very neat combination of simulations and experimental data. It is demonstrated that, under flow, cells encased by VPS largely remain in place, whereas cells in the depletion aggregates and not encased by VPS are removed by shear forces. It is suggested that in this process non-VPS producing cells are excluded. Here, it appears unclear from the data if these cells really do not produce VPS. Is it possible that VPS is just naturally distributed inhomogeneously, with some cells being covered less than others although producing as much VPS? A direct measurement of VPS production in cells could also improve the simulations since the parametrization could be improved.

Response: We wish we could use a fluorescent reporter to address this question, i.e. directly visualize the expression level of *vpsL* in the two populations in Figure 6. Unfortunately, in LB we did not obtain a good fluorescent reporter signal. We could only indirectly address this issue in the manuscript by showing that as the fraction of cells no longer surrounded by VPS decreases, the expression level of *vpsL* (one of the key VPS biogenesis genes) also decreases at the same time. This experiment was performed using luminescence measurements. We inferred that cells not surrounded by VPS (Fig. 4a) correspond to cells that no longer produce VPS – we hypothesized that these are the same expelled cells in Fig. 6, which is consistent with the observation that these cells are easily washed away by flow.

-A second comment on Fig. 6 regards speculation about the ecological function of such a process: sometimes, dispersal can be advantageous. Expelling cells that do not produce VPS may only be advantageous from the perspective of the existing colony. If daughter colonies produce less VPS, since they are offspring of “cheaters”, the process could reduce in the overall fitness of the species.

Response: Dispersal can be indeed advantageous, as the reviewer mentioned, in particular when nutrients are scarce and cells need to leave the existing biofilm and explore new territories. We also agree that these expelled cells are “cheaters” only from the perspective of the existing colony; they are fully capable of producing VPS and reestablishing new biofilm when nutrient level increases again – in fact, this is why we performed the experiment in Fig. 6f where we collected these cells and showed that they can regrow into biofilms that are indistinguishable from the original ones, in fresh media.

-The measurements on the role of alginate for depletion interaction of *P. aeruginosa* are very interesting and indeed generalize the finding for *V. cholerae*. However, it should be explained also in the main text that a *PAO1ΔpelΔpsl* is used for the experiments. This is a very particular mutant.

Response: We thank the reviewer for pointing this out; we clarify this point in the updated main text and SI figure legend and gave justification for why we used this mutant. Essentially, we want to avoid confounding factors related to the self-production of matrix in the aggregation assay when we mixed cells with matrix polymers. The *PAO1ΔpelΔpsl* mutant does not produce any known exopolysaccharide matrix and therefore was used in this assay.

- A couple of typos: Course-grained simulation -> Coarse ...

Response: Done. We have also carefully checked the entire manuscript for typos.

Reviewer #3:

This is an interesting study that focusing on analysing the cell-matrix interactions during biofilm-aggregate formation, using the model organism *Vc*. Authors present some interesting results; I particularly like the phase diagram analyses. However, my main concern is that in the current form, a number of conclusions are not supported by the data, as presented. My specific comments are below.

Response: We sincerely thank the reviewers for the extensive, constructive feedback. In the updated manuscript, we have strived to clarify our results by performing additional experiments and better explaining the rational for our experiments. Please see our point-by-point responses below. We hope the manuscript is more solid after integration of these new results.

1. As authors are most likely aware, the biofilm field is shifting to analyse biofilms in models that more closely represent the natural environments that these organisms encounter. This study is highly artificial, both in the media used to grow the biofilm-aggregates (LB or M9; this was unclear), and that only mutants were analysed. It is unclear if these observations can be replicated with WT cells, or in nutritional conditions that mimic *in vivo* environments. It is therefore unclear how relevant these observations are to the ecology of the organism, outside of this artificial lab conditions.

Response: We thank the reviewer for pointing out this trend in biofilm studies. Indeed, we have ongoing and published work about biofilms in the native context, i.e. in animal model for *V. cholerae* (*Vc*) (see for example PMID:35343438). However, we think that controlled experiments in “artificial lab conditions” still provide critical information regarding biofilm formation, in the absence of complex host factors. Particularly, when it comes to studying the biochemistry of the matrix or the biophysics involved in biofilm growth, it is important to have controlled conditions that can be easily manipulated in the lab. And such *in vitro* results can be later validated *in vivo*.

The use of the high c-di-GMP cells allowed us to focus on the biophysical aspects of biofilm growth independent of environmental sensing through c-di-GMP. *Vc* cells have over 60 enzymes that modulate the intracellular c-di-GMP level in response to various environmental signals; such responses interfere with our focus on cell-matrix interactions so we decided to use a locked mutant. This practice is also common in the field; see Berk *et al. Science* 2012, Yan *et al. PNAS* 2016, Patapova *et al. mBio* 2024, Ohmura *et al. Adv. Mater.* 2024, just to name a few. In response to the reviewer’s request, we performed a subset of experiments in the wild-type background. In the updated Figure S14, we showed that the ΔABC mutant in the wild-type background can show similar depletion-aggregation phenomenon, in a confined geometry. See more details in our response to question 7 below.

The biofilm matrix of *Vc* is also highly complex, including at least VPS and several matrix proteins. To disentangle the contribution of each component, we decided to do “addition” rather than “subtraction”: we started from the $\Delta 5$ strain that produces no major matrix components, and added back matrix components (one or two at a time) by using mutants with fewer genes knocked out.

2. Authors make a lot of claims regarding VPS configurations in the matrix, i.e VPS coating Vc cells, being crosslinked by Bap1. However, currently these conclusions are not supported by the data, either due to a lack of quantification or microscopy images that are unclear. Imaging VPS in the matrix (with cells) by super resolution microscopy would support these conclusions.

Response: Super-resolution imaging of VPS in the matrix has been done; see *Berk et al. Science* 2012. However, the dynamic changes in cell-matrix interactions and the matrix organization are difficult to study with super-resolution microscopy, which requires a much more stringent imaging condition often requiring fixed samples. We have attempted to use cryoEM to visualize VPS in the native state in a biofilm, but the contrast of VPS in cryoEM is too low to generate reliable results. Therefore, we decided to take a holistic approach combining bacterial genetics, microscopy, biochemistry, and computer simulations to probe the cell-matrix interaction. Each individual experiment or assay only yields partial information about the system, but together they advance our understanding of the matrix. Finally, in the updated Fig. 3c, we provide clear microscopic images showing VPS signals surrounding cells actively secreting VPS.

3. Due to the mutants used across the different assays it is difficult to compare results. $\Delta 5$ doesn't produce matrix components, ΔABC doesn't produce matrix proteins, but overproduces VPS, while $\Delta rbmA\Delta rbmB$ produces WT amounts of VPS. This complicates interpretation.

Response: The use of different mutants in different assays is intended to disentangle the interactions between cells and VPS. Given the complexity of the biofilm matrix and consequently cell-matrix interactions, we had to use different mutants to test each separate hypothesis. This complexity is inherent to biofilm studies, and that is why, even long after these genes are discovered, we still do not have a clear biophysical understanding of the biofilm formation process. We acknowledge that comparing results across mutants can be challenging; we tried our best in the updated manuscript to clarify why we use each mutant for each assay.

4. Authors may want to consider determining if the $\Delta 5$ mutant has the same surface change, LPS structure etc compared to WT. This would be an important control as a number of surface structures have been deleted which may introduce artifacts into the results.

Response: We appreciate the reviewer's question about the surface property of the cells. In response to the reviewer's question, we performed two characterizations of the surface properties of fixed 5Δ and ΔABC cells in growth phase I and II: a) δ -potential (surface charge); b) surface hydrophobicity, by using the microbial adhesion to hydrocarbon (MATH) assay with two different hydrocarbons (xylene and octane) (PMID:16923066). We did not observe substantial change in either surface charge or hydrophobicity for these three types of cells.

We agree that we have not extensively characterized other surface properties such as LPS structures, which will require substantially more work and beyond the scope of the current study. Because the VPS secretion machinery is not known to affect any other cell surface structures, we think that the presence or absence of VPS is therefore the dominant difference in cell surface that controls aggregation behavior presented in the current study.

Figure R1. Cell surface properties do not change significantly during biofilm growth. (a) δ -potential as a measure of surface charge, for 5 Δ cells and ΔABC cells fixed at 8 or 20 hours. *Vc* cells have a δ -potential around -50 mV consistent with literature value (PMID:33932437), and this value does not change significantly during biofilm growth (compare 8h to 20h for ΔABC cells) or upon deletion of *vpsL*. δ -potential was measured using Horiba Nano particle analyzer SZ-100V2 with an electrode cell. (b) Results from microbial adhesion to hydrocarbon (MATH) assay show that hydrophobicity of cells does not change significantly during biofilm growth or upon deletion of *vpsL*. In short, 1 mL of cell culture (washed and resuspended in PBS) was mixed with either xylene (left) or octane (right), vortexed, and allowed to separate into layers. The OD₆₀₀ was measured after and before the treatment and the ratio was taken as an indication for the preference of the *Vc* cells to remain in the aqueous solution. MATH assay is widely used as a measure of the hydrophobicity of bacterial cells (PMID:16923066).

5. Figure 1a. Are the top two fluorescent images zoomed insets of the lower image? Is the top image of $\Delta 5$ incubated with chitosan and middle image of $\Delta 5$ incubated with pVPS? What is the lower image? Clarifying this in the legend would be helpful, as this was not clear.

Response: We apologize for the confusion. We updated the figure caption in 1a to explicitly clarify these points and provide a more detailed description to ensure that the context is clear. Specifically, they are images taken in different polymer solutions: chitosan (top), purified VPS (middle), and no polymer as a control (bottom).

6. Figure 1a and S2. Assuming that the CFU value on the x-axis is the amount of bacteria added to the polymer? Or was this the amount after the 6h incubation? This was not clear.

Response: To clarify, the CFU/mL values on the x-axis represent the quantity of bacteria added to the polymer solution at the beginning of the aggregation assay. All experiments for generating phase diagrams were conducted in M9 medium without any carbon source, which means there was no bacterial growth during the incubation period. Therefore, the CFU/mL values also reflect the quantity of bacteria after the 6-hour incubation.

7. Figure 1b. Do authors see the same phenotype in ΔABC , without the DGC mutation? It would be interesting to determine if overproduction of VPS is required for this phenotype.

Response: We have now shown that the ΔABC mutant in a wild-type background (without the DGC mutation) can show similar depletion-aggregation phenomenon, in a confined geometry (Fig. S14). In an open geometry, the VPS concentration in the bacterial culture produced by WT cells probably never reached the threshold for depletion aggregation. When grown under a 1.5% agarose gel that confines both the cells and polymers, we did observe spontaneously formed depletion aggregates with morphologies similar to those cells with the DGC mutation. This experiment suggests that the biophysical principle we discussed in the paper remains valid in the WT background. Moreover, such physical confinement may also be commonly experienced by Vc cells when they grow inside the mucosal layer (see for example PMID:32355001 and 20689747).

8. Figure 1b. Authors state that depletion aggregation is occurring. However, from the representative images it is unclear what the cellular arrangements are. I would suggest providing clearer images to support this conclusion.

Response: We have provided zoom-in images of the spontaneous depletion-aggregates observed during the growth of ΔABC control at different time points, in updated Fig. S4.

9. Figure 1d. How did authors control for aggregation reducing the OD values, by the aggregates settling out of solution? Would CFU/mL be a more accurate measurement.

Response: As detailed in the method session, when measuring OD for this case, we diluted and vortexed the sample to dissociate the aggregates – the depletion-aggregates broke easily into individual cells, which we confirmed using microscopy. We did the OD measurement manually at each time point exactly because of the issue of settlement and aggregation. We have included these experimental details in the Methods session.

10. Figure 2a-c. I am confused by this data, and the conclusions drawn from it. Couldn't the differences be attributed to comparing VPS overproducing cells (ΔABC) to non-VPS producing cells ($\Delta 5$)?

Response: We apologize for the confusion. Indeed, the first point of the figure is that the interactions of VPS with VPS-overproducing cells (ΔABC) and with non-VPS producing cells ($\Delta 5$) are different, as shown by the gap between the solid and dashed lines in Fig. 2a. We further showed in Fig. 2b-c that this gap can be modulated depending on the growth phase or RbmB treatment, which further shows the importance of the cell surface state in determining cell matrix interactions. We have updated this paragraph to make it clearer to the reader.

11. L158-159. What is the evidence for this?

Response: At this point of the paper, it was a conjecture. This point will be confirmed later by VPS staining in Figure 4. We were not able to stain VPS in the ΔABC mutant, consistent with prior literature report (Berk *et. al Science* 2012; Yan *et al. PNAS* 2016) that Bap1/RbmC is required for WGA staining for VPS. The reason is unknown. We have attempted to use cryoEM tomography to visualize the surface anchored VPS, but the contrast of exopolysaccharide in EM is too low.

12. L163. Authors state that $\Delta ABC + RbmB$ have a phase diagram similar to $\Delta 5$. However in Figure 2c the solid ($\Delta ABC + RbmB$) and dashed ($\Delta 5$) lines are very different. Please clarify this statement.

Response: Thanks for pointing this out. We have updated the text to state that the phase diagram in Fig. 2c (pVPS + dABC cells in growth phase I + RbmB) more resembles the phase diagram in Fig. 2b (pVPS + dABC cells in growth phase II) than the phase diagram in Fig. 2a (pVPS + dABC cells in growth phase I), showing the effect of surface trimming by RbmB.

13. Quantifying VPS levels, or staining VPS and visualizing alone with the cells by microscopy throughout the experiments, especially for the ΔABC and $\Delta ABC + RbmB$ experiments, would help support the conclusions.

Response: We sincerely thank the reviewer for this suggestion. We were not able to stain VPS in the ΔABC mutant, both *in vitro* and in biofilm (see Q11 above). We have attempted various methods during the revision stage to directly quantify VPS in this mutant, all in vain. However, we were able to use the luminescence reporter to quantify the expression level of *vpsL* in this mutant, which shows a similar pattern as the other mutants: it peaks in the first growth phase and declines in the 2nd growth phase; see below.

Figure R2 VPS production during ΔABC biofilm growth. Quantification of *vpsL* gene expression in $\Delta rbmA \Delta bap1 \Delta rbmC$ (ΔABC) biofilms through the measurement of luminescence from the pBBRlux-*vpsL* reporter, normalized by OD₆₀₀. Data are presented as mean \pm SD ($n = 4$ biological replicates).

14. L173. Authors states that induction of VC1086 abolishes VPS production. However, there is no quantification of VPS levels. This is an overinterpretation of the data. Furthermore, the images presented in Figure 2d as evidence of loss of aggregation are difficult to interpret. Both appear to be a lawn of cells.

Response: Reference 44 has documented that the induction of VC1086 can push the intracellular concentration of c-di-GMP to a very low level ($\sim 1 \mu M$); we are using the same plasmid from this laboratory. Moreover, because extensive literature has shown that c-di-GMP level is positively correlated with VPS levels, we infer that induction of VC1086 will abolish VPS production. We have given more background in this session to explain the rationale of the experimental design.

The top image in the Fig. 2d shows a culture of freely swimming *V. cholerae* cells, and thus appearing as a lawn with homogeneous cell density. The bottom image in Fig. 2d, instead, shows irregular cell density due to depletion aggregation. Again, we have now provided zoom-in images to distinguish the two cases (Fig. S8).

15. L177 – 183. These conclusions are not supported by the data. Authors have not quantified either c-di-GMP or VPS levels in these experiments.

Response: The effect of this plasmid on c-di-GMP has been extensively characterized in Reference 44. All phenotypes in the presence or absence of IPTG we observed are also consistent with Reference 44 and known regulation of c-di-GMP on VPS.

16. L207 – 208. This experimental set up was confusing. Assuming purified VPS and Bap1 were mixed together? VPS and Bap1 alone controls need to be included. The electron microscopy images in Figure S7 are unclear.

Response: Yes, the image shows that clumps are formed when purified VPS and Bap1 were mixed together. We have included the Bap1-only and VPS-only controls in the updated Fig. S9a-b; they do not show any clumps.

Regarding the electron microscopy results, we have included additional ratios of Bap1 and VPS in Fig. S10. While VPS alone does not have enough contrast in EM, in the presence of Bap1, one can observe clumps of increasing sizes when more VPS was added. We have included the clarification in the legend of Fig. S10.

17. Figure 3b. For the phase diagrams in this panel and throughout the other figures, how did authors define an aggregate? Was there a minimum number of cells incorporated? Authors state that in this figure, no aggregates were observed. However, in the image provided there appears to be two aggregates. Also Bap1 should also be labelled to confirm the reported conclusions, that Bap1 crosslinks with VPS, preventing aggregation.

Response: We define an aggregate as a cluster of at least three cells close to one another. We agreed with the reviewer that the image and the resulting text for the original Fig. 3b are not clear; therefore, we decided to remove this panel and the corresponding text because it is not central to the conclusion in this figure and the main story.

Regarding Bap1, please see our response in the next question.

18. L218 – 228. Authors make claims regarding Bap1 localization, however Bap1 is not labelled in any of the microscopy images. Therefore, authors have over interpreted their results.

Response: It is well established from prior work (Berk *et al. Science* 2012; Huang *et al. Nat. Commun.* 2023) that Bap1 co-colonizes with VPS away from the glass substrate. In response to the reviewer's questions, we have performed new experiments to simultaneously visualize Bap1 and VPS on the surface of *Vc* cells (Fig. 3c). Indeed, they colocalize.

19. L243. From the images provided in Figure 4 it is difficult to observe the cellular arrangement that supports the claim that aggregates are formed by depletion.

Response: We provided a zoom-in view in Fig. S12.

20. Figure S8. Authors state that RbmC abolished bridging aggregation. However, in the provided image a large aggregate is present, that appears to have a clumped cellular arrangement. Can authors clarify this statement.

Response: In the original experiment in Fig. S8B (now Fig. S9d), we added RbmB to cells whose surface are VPS-coated. This releases VPS through the lyase activity of RbmB. The released VPS molecules, in turn, act as depletants to aggregate these cells, whose surfaces were now devoid of VPS. To simplify the interpretation, we added a washing step to remove these released VPS, and indeed, no aggregates were observed anymore, showing that Bap1 alone cannot bridge cells whose surfaces are devoid of VPS. We have updated our results accordingly.

21. Figure S9. The conclusion that $\Delta 5$ cells are excluded from the aggregates is unclear. This should be confirmed with imaging analysis. The fact that there appears to be few $\Delta 5$ cells further confound the results. Furthermore, L289 authors state that expelled cells form depletion aggregates. However the $\Delta 5$ cells in Figure S9 are single cells, and do not support this conclusion.

Response: We agree with the reviewer that conclusion from this figure is ambiguous. We have therefore removed this figure and leave this idea for future investigation.

22. Figure 6. What was the rationale for using Δ rbmA mutant for these experiments?

Response: We wanted to stain VPS with WGA, which is not possible in the ΔABC mutant. We did not use any *rbmA*⁺ strain in the study because we do not have access to purified RbmA to perform the mechanistic study as we did for Bap1. As a compromise, we used the Δ rbmA mutant in which VPS can be stained and the results can be compared with the corresponding *in vitro* experiment. We have included an extended discussion on the possible role of RbmA in the Discussion session.

23. L403 – 404. Authors state that aggregation assays were performed in M9, and biofilm experiments were performed in LB. However, all experiments in the manuscript focused on aggregation. It is unclear what growth conditions were used.

Response: In all biofilm growth assays, LB was used because it is one of the most common media used in the microbiology laboratories. All aggregation assays and experiments to generate phase diagram were conducted in M9 minimal medium without glucose, to avoid confounding effect of growth during the assay (which cannot be performed in LB). We have clarified this point in multiple places in the Methods session.

Reviewer #4:

The manuscript of Moreau et al. investigated bacteria-matrix interaction in *Vibrio cholera* and

suggesting a new conceptual. In general, the data and model can support the conclusion although the logic is no very easy to follow.

Response: We appreciate the reviewer's interest in our manuscript and agreeing with our conclusion. We have significantly updated the manuscript to streamline the logic. We have also provided more background information based on the reviewer's question throughout the text. We thank the reviewer for making the manuscript a better one.

1. Introduction should include more information, such as the current understanding about the function of VPS, RbmABC and Bap1 as well as their contribution in the matrix; How C-di-GMP affect the expression of these genes, etc. These will help reader to understand the results.

Response: We apologize for not including such information in our original manuscript. We have now added the relevant background information in the introduction or in appropriate places.

2. What is the function of pomA? I did not find any information about this gene in entire manuscript.

Response: PomA is part of the flagellar motor in *Vc* and is required for cell motility; deletion of *pomA* results in nonmotile cells. To make the aggregation assay easier, we use nonmotile cells in this particular assay. For biofilm growth, we didn't have to use this mutation because *Vc* cells downregulate motility in the biofilm state. We have clarified this point in the main text.

3. Is VPS a secreted exopolysaccharide or bacterial surface associated? If this is the first study about the localization of VPS, the author should provide more data.

Response: It is well established that VPS is a secreted *exopolysaccharide*; see Yildiz *et al. PNAS* 1999; Fong *et al. Microbiology* 2010; Yildiz *et al. PLoS One* 2014; Huang *et al. Nat. Commun.* 2023, etc. However, the main point of our paper is that as VPS is being secreted, it is temporarily anchored on the cell surface – and this matters for cell-matrix interaction.

4. L118, How the *vpvC* mutant affect the C-di-GMP level, increase or decrease?

Response: The *vpvC* mutant has an elevated c-di-GMP level.

5. L238, Why use *rbmA* deletion mutant? If in a RbmA⁺ background, what would be the results?

Response: We do not have access to purified RbmA to perform the same mechanistic studies as we have shown here for Bap1 (and RbmC), so we have not included RbmA⁺ strains in our study. The RbmA⁺ strains also undergo dispersal under the same condition (data not shown) and therefore, similar inversion of cell-matrix interaction potentially. There is no depletion-aggregation in the RbmA⁺ culture even in the late stage, for reasons we do not fully understand yet. We have discussed the potential role of RbmA in the Discussion session.

6. L259, Does RbmB release VPS or degrade it?

Response: This is an excellent question and is hard to answer. We have a separate manuscript (doi.org/10.1101/2024.08.27.609776) regarding the lyase activity of RbmB, in which we also showed that RbmB can enzymatically cleave VPS *in vitro*; its mode of action in biofilm is still unclear. Our data suggests that it can trim off the surface-anchored VPS and release it, which is consistent with its enzymatic activity. Whether RbmB is secreted and whether it degrades the VPS network in a biofilm is currently unknown, and our updated manuscript includes a dedicated discussion on this point.

7. Fig.2, need the control of vector –IPTG for 13h,+IPTGfor 12h

Response: Done. We added the control to Fig. S8d.

8. Fig.4, How to determine the phase I and II? There is a strong signal of VPS at 4.5h, but little at 13h. Where those VPS has gone? Be degraded or removed by staining processing?

Response: We used two criteria to determine phase I and II, as shown in Fig. 4b: 1) sharp increase of cell number after the first plateau in the grow curve; 2) the beginning of the sharp decrease of fraction of cells encased by VPS.

There are a couple of potential sources of the decay of VPS staining signal. 1) Photobleaching due to continuous imaging; 2) Degradation or release of VPS by RbmB; 3) Lowering of available WGA molecules for staining in the imaging chamber. We preferred the second explanation because in Fig. 5A, the corresponding $\Delta rbmB$ biofilm does not show decay in the signal encasing the biofilm cluster. We have updated the text accordingly.

9. For *Pseudomonas aeruginosa*, Psl polysaccharide is more important for biofilm formation and cell-cell interaction. Mucoid strain also requires Psl to form biofilm. I am wondering whether the depletion-aggregation occurs in PAO1.

Response: We did not see similar depletion aggregation in the PAO1 strain nor the PA14 strain. The mechanism of how Psl or Pel forms the biofilm matrix seems to be very different from alginate; further investigation is warranted along this line. We thank the reviewer for pointing out that mucoid strain also requires Psl to form biofilms (PMID:28634241 and 22309122). For data shown in the current Fig. S15a, we were using a mucoid strain ($\Delta mucA$) in which alginate was suggested to be the dominant matrix component. For the aggregation assay in the current Fig. S15b, we want to avoid confounding factors related to the self-production of matrix in the aggregation assay when we mixed cells with alginate, so we used the PAO1 $\Delta pel\Delta psl$ mutant, which does not produce any known exopolysaccharide matrix. Therefore, there is a possibility that spontaneous depletion aggregation we saw in the mucoid strain may be related to the degradation of Psl during biofilm growth – future experiments are needed to test this interesting hypothesis. Due to various unknowns in the *Pa* system, we have decided to merge the *Pa* session with Discussion (instead of a standalone session) in the updated manuscript.

Reviewer #5:

Response: We appreciate your time and efforts in reviewing our manuscript.